# The proteostatic landscape of healthy human oocytes

Gabriele Zaffagnini [ID][1,6], Miquel Solé [ID][2], Juan Manuel Duran[1], Nikolaos P Polyzos[2,3] & Elvan Böke [ID][1,4,5][✉]

## Abstract

Oocytes, female germ cells that develop into eggs, are among the longest-lived cells in the animal body. Recent studies on mouse oocytes highlight unique adaptations in protein homeostasis (proteostasis) within these cells. However, the mechanisms of proteostasis in human oocytes remain virtually unstudied. We present the first large-scale study of proteostatic activity in human oocytes using over 100 freshly donated oocytes from 21 healthy women aged 19–34 years. We analysed the activity and distribution of lysosomes, proteasomes, and mitochondria in both immature and mature oocytes. Notably, human oocytes exhibit nearly twofold lower proteolytic activity than surrounding somatic cells, with further decreases as oocytes mature. Oocyte maturation is also coupled with lysosomal exocytosis and a decrease in mitochondrial membrane potential. We propose that reduced organelle activity preserves key cellular components critical for early embryonic development during the prolonged maturation of human oocytes. Our findings highlight the distinctive biology of human oocytes and the need to investigate human-specific reproductive biology to address challenges in female fertility.

**Keywords** Female Fertility; Human Oocytes; Proteostasis; Lysosomes; Mitochondria
**Subject Categories** Development; Organelles; Post-translational Modifications & Proteolysis

## Introduction

A remarkable feature of reproduction is that offspring do not inherit the ageing status of their parents, even though the embryo inherits its cytoplasm from the oocyte. This suggests the existence of mechanisms that counteract the accumulation of intracellular damage in oocytes. Protein degradation plays a central role in maintaining protein homeostasis (proteostasis) in all cell types. It is facilitated by two major cellular waste disposal systems: the ubiquitin–proteasome system and the autophagy–lysosomal pathway. Soluble proteins are typically degraded by the ubiquitin-proteasome system, whereas bulkier intracellular materials, such as protein aggregates and damaged organelles, are targeted by autophagy and degraded in lysosomes (Pohl and Dikic, 2019). Unsurprisingly, impairment of protein degradation machinery has been linked to defects across all stages of human life, from altered embryogenesis to ageing (López-Otín et al, 2023; Allen and Baehrecke, 2020). However, these mechanisms remain virtually uncharacterized in human oocytes.

Oocytes are formed during fetal development and can remain dormant in the ovary for decades. This long lifespan requires oocytes to preserve their cytoplasmic integrity over time. Recent evidence from studies in mice highlights unique adaptations in oocytes for maintaining proteostasis: first, mouse oocytes exhibit exceptionally long protein half-lives, suggesting that their longevity may be tied to the durability of their proteome (Bomba-Warczak et al, 2024; Jentoft et al, 2023; Harasimov et al, 2024). Second, mouse oocytes contain specialized compartments, called ELVAs, for storing and degrading protein aggregates (Zaffagnini et al, 2024; Satouh et al, 2024). These adaptations underscore the critical role of proteostasis in safeguarding oocyte quality and functionality in mice. Based on mouse studies, one might expect that human oocyte maturation is coupled to increased lysosomal and proteasomal activity. Here, we set out to test this assumption directly.

Despite extensive evidence linking the loss of proteostasis to ageing in other cell types, little is known about the proteostatic machinery in human oocytes, making it challenging to investigate age-related declines in proteostatic mechanisms (Duncan et al, 2017; Galatidou et al, 2024). To address this gap, we conducted the first systematic survey of organelle activity and distribution in live, healthy human oocytes at different stages: immature germinal vesicle (GV) oocytes and fertilization-ready metaphase II (MII) eggs. We did not use in vitro matured (IVM) oocytes in this study as IVM leads to suboptimal fertility outcomes (Das and Son, 2023). Using a substantial sample size—over 70 MIIs and 30 GVs collected directly from 21 healthy donors—our study provides a comprehensive characterization of the proteostatic machinery in human oocytes.

## Results

### GV-stage oocytes have higher degradative activity than MIIs

Decreased levels of proteostasis proteins have recently been correlated with poor outcomes in aged human oocytes (Galatidou

[1]Centre for Genomic Regulation (CRG), The Barcelona Institute of Science and Technology, Barcelona, Spain. [2]Dexeus Fertility, Department of Obstetrics Gynecology and Reproductive Medicine, Hospital Universitari Dexeus, Barcelona, Spain. [3]Faculty of Medicine and Health Sciences, University of Ghent, Ghent, Belgium. [4]Universitat Pompeu Fabra (UPF), Barcelona, Spain. [5]Institució Catalana de Recerca i Estudis Avançats (ICREA), Barcelona, Spain. [6]Present address: CECAD Research Centre, Faculty of Mathematics and Natural Sciences, University of Cologne, Cologne, Germany. ✉E-mail: Elvan.boke@crg.eu

et al, 2024); however, there is virtually no evidence regarding the expected, or normal, levels of proteostatic activity in healthy human oocytes. To address this gap, we set out to characterize the activity of protein degradation machineries in healthy human oocytes. For this, we analysed immature (GV-stage) and mature (MII-stage) oocytes retrieved from healthy donors undergoing oocyte donation cycles for heterologous in vitro fertilization (Table EV1). As expected (Pors et al, 2022), GVs were on average slightly smaller than MIIs (Fig. EV1A). The chromatin configuration of the retrieved GVs was consistent with previous reports in the literature (Combelles et al, 2002) (Fig. EV1B,C). MIIs were randomly selected after collection and were morphologically indistinguishable from sibling oocytes used for subsequent IVF.

We began our study by probing both GV and MII oocytes with LysoTracker and Me4BodipyFL-Ahx3Leu3VS (Me4Bpy) to assess the activity of lysosomes and proteasomes, respectively (Fig. 1A). LysoTracker accumulates in active lysosomes based on acidic pH (Chazotte, 2011), whereas Me4Bpy marks active proteasome cores (Berkers et al, 2007). To serve as an internal control, we left some cumulus cells attached to the zona pellucida of the oocytes, enabling comparison between each oocyte and its surrounding somatic compartment. Both the oocytes and the cumulus cells remained alive without major stress under our imaging conditions, as determined by their stable mitochondrial membrane potential measured with TMRE (Figs. 1A and EV1D). TMRE is a dye sensitive to the mitochondrial membrane potential, and it was shown to be less prone to artefacts than the ratiometric probe JC-1 in mouse oocytes (AL-Zubaidi et al, 2019). In cumulus cells, both LysoTracker and TMRE showed labelling patterns consistent with the expected localization and morphology of lysosomes and mitochondria, respectively (Fig. EV1D). Both nuclear and cytoplasmic Me4Bpy accumulation could be detected in cumulus cells, as expected for active proteasomes in somatic cells (Fig. EV1D) (Enenkel et al, 2022). In addition, the labelling of each dye was suppressed in both oocytes and cumulus cells by a specific inhibitor of the corresponding organelle's activity, confirming that these live dyes specifically label their respective compartments (Fig. EV1E–G).

We then investigated the activity of lysosomes and proteasomes in oocytes, using mitochondrial activity as an internal reference, given that it is better characterized than degradative pathways in oocytes. For this, we compared LysoTracker, TMRE and Me4Bpy intensities in GV- and MII-stage oocytes retrieved from the same donors. MIIs accumulated less LysoTracker compared to their sibling GVs, indicating lower lysosomal activity (Figs. 1A,B and EV2A). Similarly, TMRE intensity was also significantly lower in MIIs than in sibling GVs (Figs. 1A,B and EV2A). Strikingly, we observed no specific Me4Bpy accumulation in either oocyte stage, and the only detected signal corresponded to autofluorescence of refractile bodies (Fig. 1A and EV2B). In contrast, no significant differences were observed in LysoTracker or TMRE intensities when comparing cumulus cells from GV- and MII-stage cumulus–oocyte complexes (COCs) within the same donors (Figs. 1A and EV2C). However, cumulus cells associated with MII-stage oocytes accumulated more Me4Bpy than those from GV-stage oocytes, suggesting increased proteasomal degradation in the somatic compartment during oocyte maturation (Figs. 1A and EV2C).

Lack of Me4Bpy accumulation in oocytes could be due to undetectably low proteasomal activity, or to inefficient retention of Me4Bpy inside oocytes (Strouse et al, 2013). To differentiate between these possibilities, we repeated labelling with Me4Bpy in presence of Verapamil, an inhibitor of ABC-family transporters, that are typically involved in multidrug resistance in cancer cells (Wu et al, 2008). Verapamil treatment increased labelling of Me4Bpy, as well as of LysoTracker and TMRE, in both oocytes and cumulus cells, indicating that these dyes are indeed exported from the cell by the activity of ABC transporters (Fig. EV2D–F) (Strouse et al, 2013; Zhitomirsky et al, 2018). The intensity increase upon Verapamil treatment was typically less pronounced in cumulus cells than in oocytes, suggesting that oocytes have higher activity of ABC transporters than the surrounding somatic cells (Fig. EV2E,F). In verapamil-treated GV oocytes, Me4Bpy accumulated in the nucleus as well as in the cytoplasm (Figs. 1C and EV2D), consistent with the expected proteasome localization in cells (Enenkel et al, 2022). Proteasomal activity was lower in MIIs than in GVs obtained from the same donors (Figs. 1C,D and EV2G).

Thus, we concluded that lysosomal, mitochondrial and proteasomal activities decrease in human oocytes during maturation.

## Human oocytes exhibit lower degradative activity compared to somatic cells

We next asked whether human oocytes have comparable degradative activity to somatic cells, by analysing each oocyte alongside its surrounding cumulus cells. For this, we measured the intensities of LysoTracker, TMRE and Me4Bpy in both cell types at the same confocal sections (Figs. 1E–G and EV2H–J). In untreated COCs, LysoTracker, TMRE and Me4Bpy intensities were lower in oocytes compared to the adjacent cumulus cells, both in GVs and MIIs (Figs. 1A,E–G and EV2H–J). Following Verapamil treatment, the intensities of LysoTracker, TMRE, and Me4Bpy remained lower in oocytes compared to their corresponding cumulus cells (Figs. 1G and EV2J,K).

Therefore, we concluded that oocytes have lower lysosomal, proteasomal, and mitochondrial activities compared to the surrounding cumulus cells.

## Lysosomal abundance decreases during oocyte maturation

We next asked whether the decrease in lysosomal and proteasomal activities from GVs to MIIs is due to a decrease in the protein levels of the degradative machineries during human oocyte maturation. We first investigated the proteasomal subunit levels in GV and MII oocytes and performed immunocytochemistry for the core 20S proteasome subunits (Fig. 2A). The immunolabelling of the 20S core proteasome displayed similar levels between GVs and MIIs, albeit with a trend towards reduction in MII oocytes (Figs. 2A,B and EV3A). This was supported by the reanalysis of the proteomics data from a recent study (Galatidou et al, 2024) (Fig. EV3B). Conversely, 20S proteasome intensity increased in cumulus cells from GVs to MIIs (Figs. 2A,C and EV3C), suggesting that cumulus cells surrounding MII oocytes have higher proteasome levels, in line with our finding that Me4Bpy labelling increases in cumulus cells during oocyte maturation (Fig. EV2C).

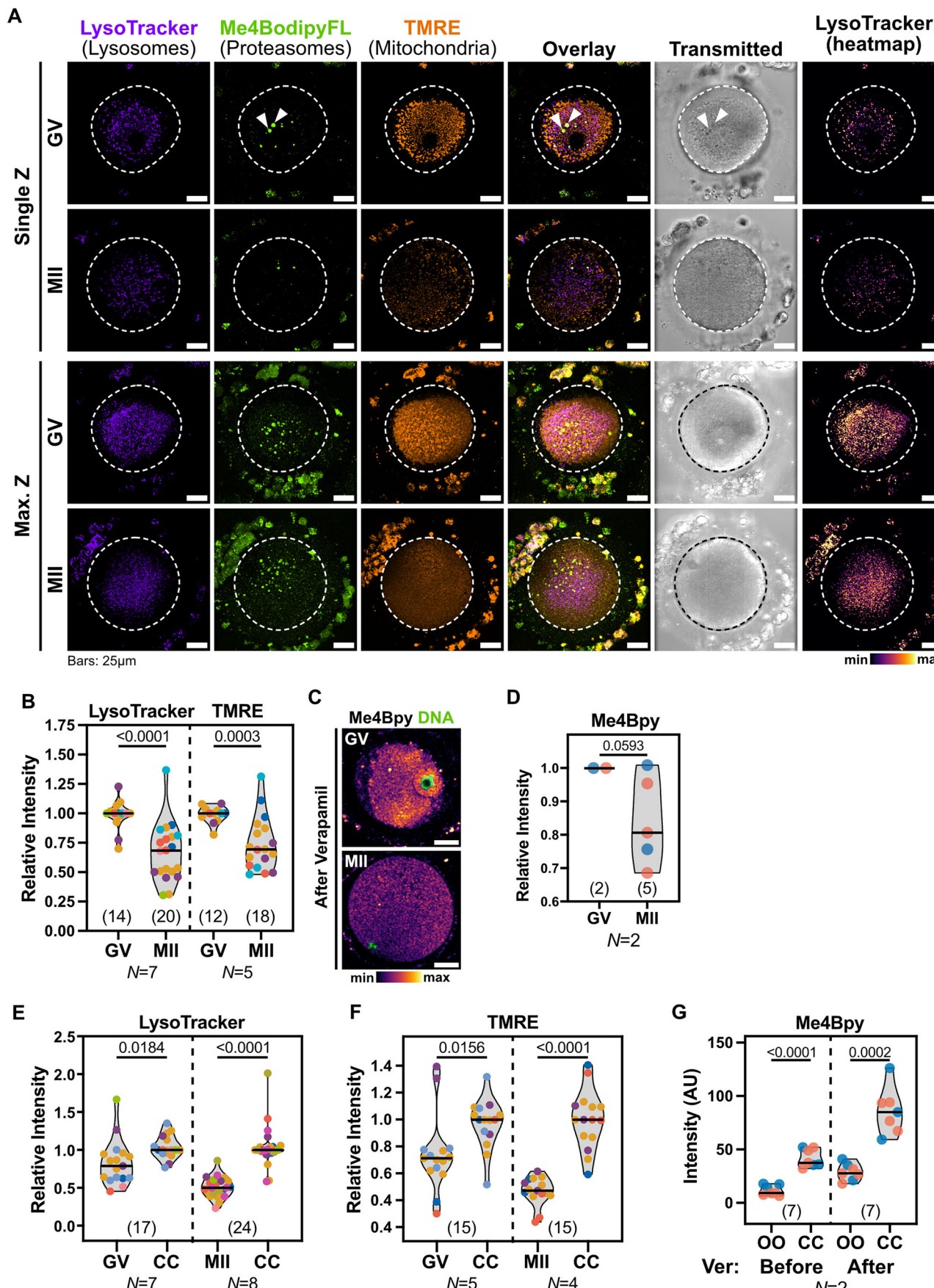

**Figure 1. Degradative and mitochondrial activity decreases during oocyte maturation in humans.**

(A) Representative live confocal images of human immature (GV) and mature (MII) oocytes labelled with LysoTracker Deep Red, Me4Bpy and TMRE to mark the activity of lysosomes, proteasomes, and mitochondria, respectively. Single confocal slices at the equatorial plane and maximal Z projections are shown. Dashed lines indicate the oocyte perimeter. Notice the clustered appearance of mitochondria and lysosomes in GVs, that leave an organelle-free area below the cell cortex. Lysosomes were also centrally distributed in MIIs. Arrowheads denote autofluorescent refractile bodies in the cytoplasm of oocytes. (B) Comparison of the average Lysotracker (lysosomes) and TMRE (mitochondria) intensity in GV and MII oocytes. Data points represent individual oocytes and are colour-coded by donor. Colour-coding is consistent across all figures (see Table EV1). Only donors with at least one GV and one MII were considered. N numbers indicate the total donors considered for each quantification. For each donor, data were normalized to the median of GVs. Numbers in parentheses indicate the total oocytes analysed per condition. Data averaged by donor are shown in Fig. EV2A. p values: unpaired t tests with Welch's correction. (C) Live Confocal images of GV and MII oocytes treated with Verapamil and labelled with Me4Bpy to mark proteasomal activity (shown as a heatmap). Scale bars: 25 μm. (D) Quantification of the average Me4Bpy intensity in Verapamil-treated GV and MII oocytes from $N = 2$ donors. For each donor, data were normalized to GVs. Data averaged by donor are shown in Fig. EV2G. p value: unpaired t test with Welch's correction. (E, F) Comparison of Lysotracker (E) and TMRE (F) intensities between oocytes and the corresponding cumulus cells (CC). For each donor, data were normalized to the median of CCs. A pairwise comparison between each oocyte and the corresponding cumulus cells is shown in Fig. EV2H,I, respectively. p values: unpaired t tests with Welch's correction. (G) Quantification of Me4Bpy intensity in the same oocytes (OO) (2 GVs and 5 MIIs from $N = 2$ donors) and cumulus cells (CC) before and after incubation with Verapamil (Ver). Comparison between GV and MII oocytes is shown in Fig. 1D. p values: unpaired t tests with Welch's correction. Source data are available online for this figure.

We then asked whether the lower lysosomal activity we observed in MII-stage oocytes was also due to reduced abundance of lysosomes in MIIs. For this, we immunolabelled the lysosomal integral membrane protein LAMP1 and the lysosomal protease Cathepsin D (CTSD) in human GVs and MIIs. Confirming our live imaging results (Fig. EV3B), the abundance of LAMP1/CTSD-positive lysosomes was strongly diminished in MIIs compared to their sibling GVs (Figs. 2D,E and EV3D). Strikingly, the reduction of intracellular lysosomes was accompanied by an increase of LAMP1 in the plasma membrane (PM) of MIIs (Figs. 2D,F and EV3E). To confirm this result, we immunolabelled non-permeabilized oocytes with an antibody (H4A3) against the luminal domain of human LAMP1 (Reddy et al, 2001), which would be exposed to the extracellular space upon insertion of LAMP1 into the PM. Surface LAMP1 was still detected in non-permeabilized MIIs (Figs. 2G and EV3F), indicating that its luminal domain is exposed to the extracellular space.

We conclude that degradative activity decreases during human oocyte maturation, due to decreased abundance of lysosomes and possibly proteasomes, in MIIs compared to GVs (Figs. 1A–D, 2A,B,D,E, and EV3A,B,D). We further conclude that the reduction in lysosomal abundance is likely due to increased lysosomal exocytosis, as indicated by the appearance of surface LAMP1 in MIIs (Figs. 2D–G and EV3E,F).

## Human oocytes contain few large aggregated protein compartments

We have previously shown that mouse oocytes accumulate aggregated proteins inside endolysosomal vesicular assemblies that we named ELVAs (Zaffagnini et al, 2024). Considering that human oocytes have lower degradative activity than somatic cells (Fig. 1A,E–G), we asked whether they would also accumulate protein aggregates. We thus incubated GVs and MIIs with the Proteostat dye, which labels protein aggregates (Leeman et al, 2018). Interestingly, in all human oocytes tested Proteostat labelled the refractile bodies (Fig. 3A), which have been suggested to be lysosomes based on their ultrastructure (Trebichalská et al, 2020; Tatíčková et al, 2023; Sathananthan, 1994). We tested this hypothesis by co-immunolabelling LAMP1 along with Proteostat. Indeed, LAMP1 typically marked the rim of refractile bodies in human oocytes, confirming their lysosomal origin (Fig. 3A). LAMP1- and Proteostat-positive refractile bodies were found in all tested MIIs from multiple donors (Fig. 3B). The overall Proteostat

intensity in refractile bodies did not significantly change between GVs and the sibling MIIs (Fig. EV4). We conclude that, rather than ELVAs, human oocytes contain aggregated proteins within a few enlarged lysosomes distributed throughout their cytoplasm.

## Lysosomes and proteasomes are asymmetrically distributed in human oocytes

Our live imaging revealed that most organelles are unevenly distributed in the cytoplasm of both GVs and MIIs (Fig. 1A). We therefore set out to quantify the spatial distribution of active lysosomes and proteasomes in human oocytes, and used mitochondria, an organelle already known to be asymmetrically distributed in GVs (Trebichalská et al, 2020; Pires-Luís et al, 2016), for comparison.

First, we plotted the average radial lysotracker and TMRE intensities in the equatorial region of GVs and MIIs, to reveal lysosomal and mitochondrial distribution in oocytes, respectively (Fig. 4A–C). Mitochondria were clustered around the nucleus in GVs, leaving an empty area below the cell cortex, whereas they redistributed throughout the entire cytoplasm in MIIs (Figs. 1A and 4A–C). In addition, TMRE accumulation was not homogeneous throughout the perinuclear cluster of mitochondria in GVs. Instead, mitochondria with the highest membrane potential were localized at the edges of the cluster (Fig. 4A–C). Similar to TMRE, LysoTracker signal was also highly clustered in GVs, with most active lysosomes interspersed within the mito-chondrial cluster (Figs. 1A and 4A–C). Unlike mitochondria, active lysosomes also remained clustered in the centre of MII oocytes (Figs. 1A and 4A–C). The distribution of both TMRE and LysoTracker remained consistent after Verapamil treatment, indicating that the observed pattern was not due to insufficient dye penetration into the cytoplasm of oocytes (Fig. EV5A).

To confirm these results, we assessed the distribution of lysosomes and mitochondria in fixed oocytes using well-established markers for these organelles (Kühlbrandt, 2019; Zhang et al, 2023). In line with our live imaging results and consistent with previous reports (Santos et al, 2024; Pires-Luís et al, 2016), LAMP1-positive lysosomes were clustered around the nucleus in GVs, and remained clustered in the centre of MIIs (Fig. EV5B–D, see also Fig. 2D). We then tested mitochondria, and immunolabelled the mitochondrial ATP synthase subunit ATP5A. Confirming our live imaging data, and as previously described (Pires-Luís et al, 2016; Trebichalská et al, 2020; Santos et al, 2024; Szollosi et al, 1986;

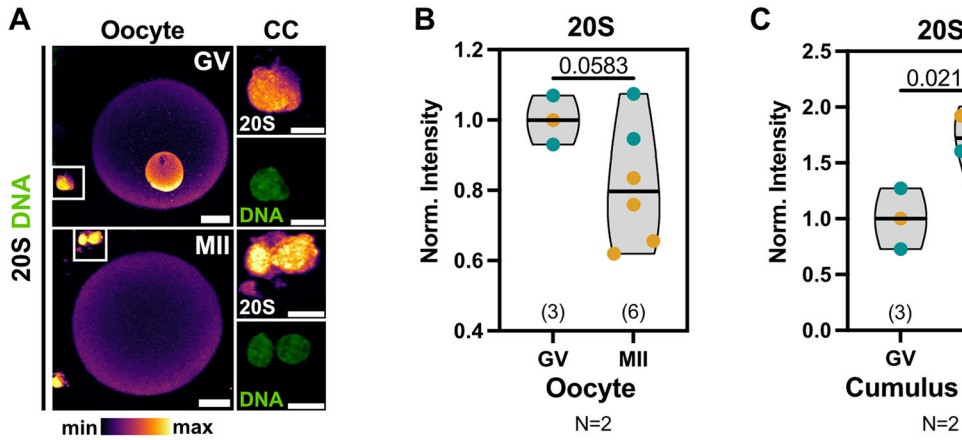

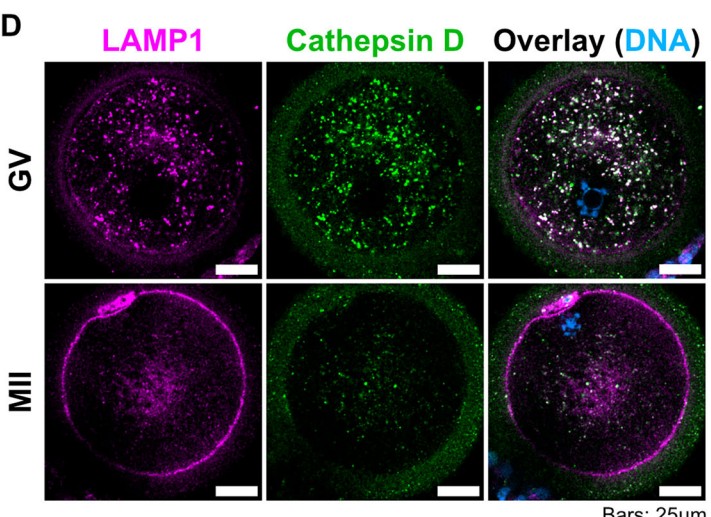

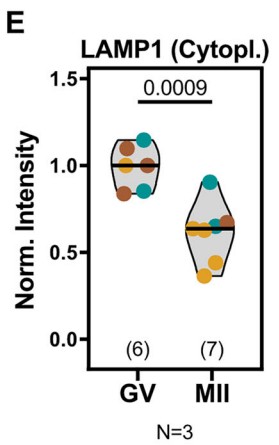

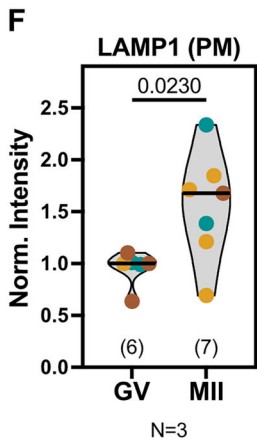

Bars: 25µm

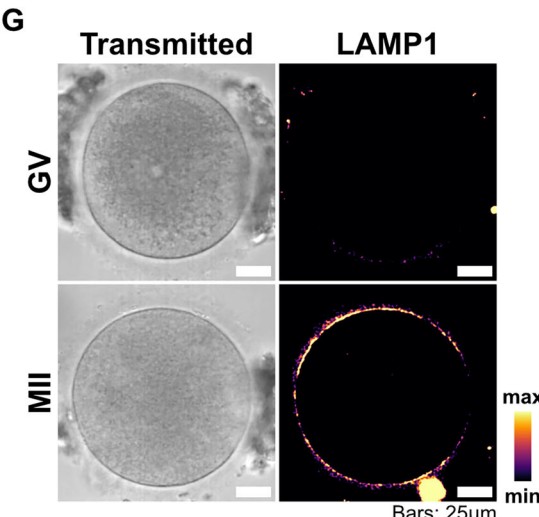

Unpermeabilized

**Figure 2. The abundance of degradative organelles decreases during oocyte maturation.**

(A) Confocal images of oocytes and cumulus cells (CC) immunolabelled with an antibody against 20S core proteasome (displayed as a heatmap). Scale bars: 25 μm. Insets: 10 μm. (B, C) Comparison of the average 20S intensity in immunolabelled oocytes (B) and cumulus cells (C) from $N = 2$ donors. Data points represent individual oocytes and are colour-coded by donor. For each donor, data were normalized to the median of GVs. The number of oocytes quantified per condition is indicated into parentheses. A pairwise comparison averaged by donor is shown in Fig. EV3A,C, respectively. $p$ values: unpaired $t$ tests with Welch's correction. (D) Representative confocal images of GV and MII oocytes immunolabelled for the lysosomal proteins LAMP1 and Cathepsin D. (E, F) Quantification of the mean intracellular (E) or plasma membrane (F) LAMP1 intensity in GV and MII oocytes shown in (D). Data points represent individual oocytes from $N = 3$ donors and are colour-coded by donor. For each donor, data were normalized to the median of GVs. The number of oocytes quantified per condition is indicated into parentheses. A pairwise comparison averaged by donor is shown in Fig. EV3D,E, respectively. $p$ values: unpaired $t$ tests with Welch's correction. (G) Confocal images of non-permeabilized GV and MII oocytes immunolabelled with anti-LAMP1 (heatmap). Scale bars: 25 μm. Source data are available online for this figure.

## A

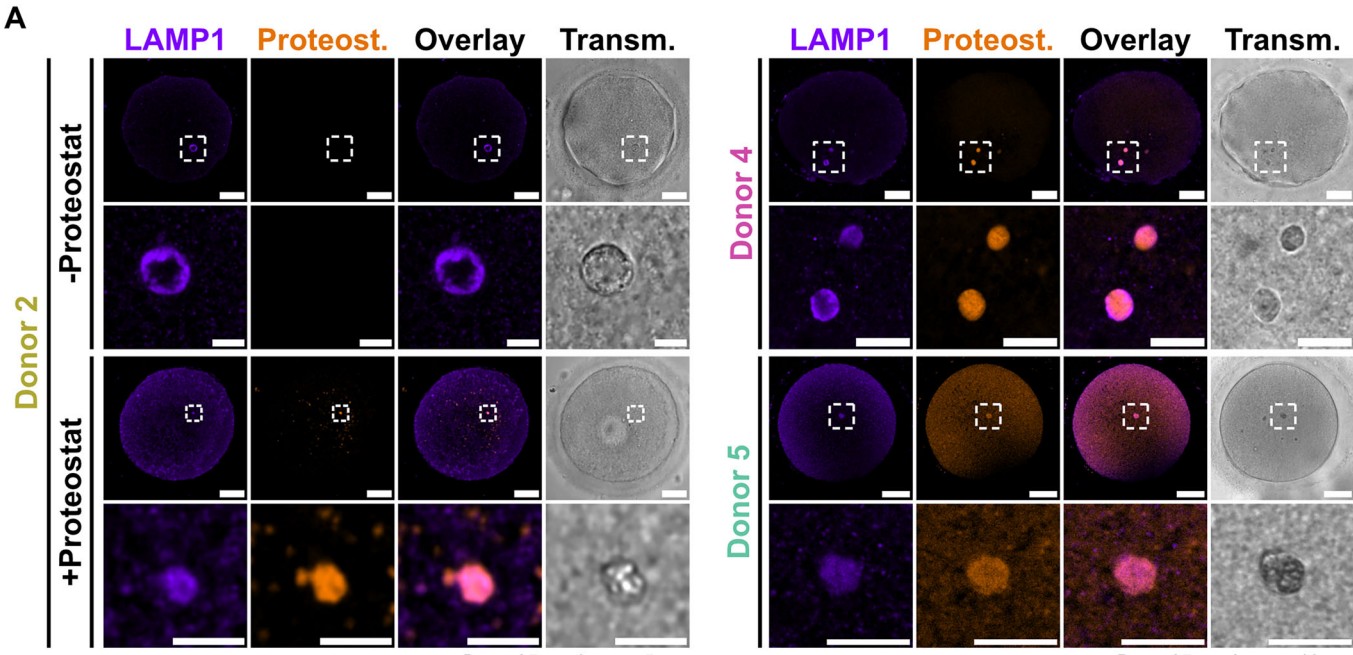

Bars: 25μm. Insets: 5μm.

Bars: 25μm. Insets: 10μm.

## B

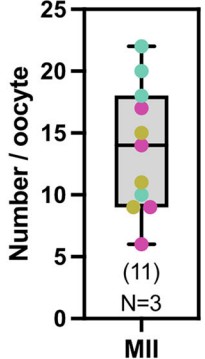

Refractile bodies

**Figure 3. Human oocytes contain aggregated proteins in few enlarged lysosomes.**

(A) Representative confocal images of oocytes from multiple donors labelled with anti-LAMP1 and with or without Proteostat. Notice the higher overall background of both LAMP1 and Proteostat in oocytes from Donor 5, highlighting variability between different donors. (B) Box-and-whiskers plot of the number of LAMP1- and Proteostat-positive refractile bodies in human MII oocytes from different donors. Data points represent individual oocytes colour-coded by donor. The horizontal line indicates the median, the box represents the interquartile range (IQR), and the whiskers denote the minimum and maximum, respectively. Numbers in parentheses indicate the total amount of oocytes quantified from $N = 3$ donors. Source data are available online for this figure.

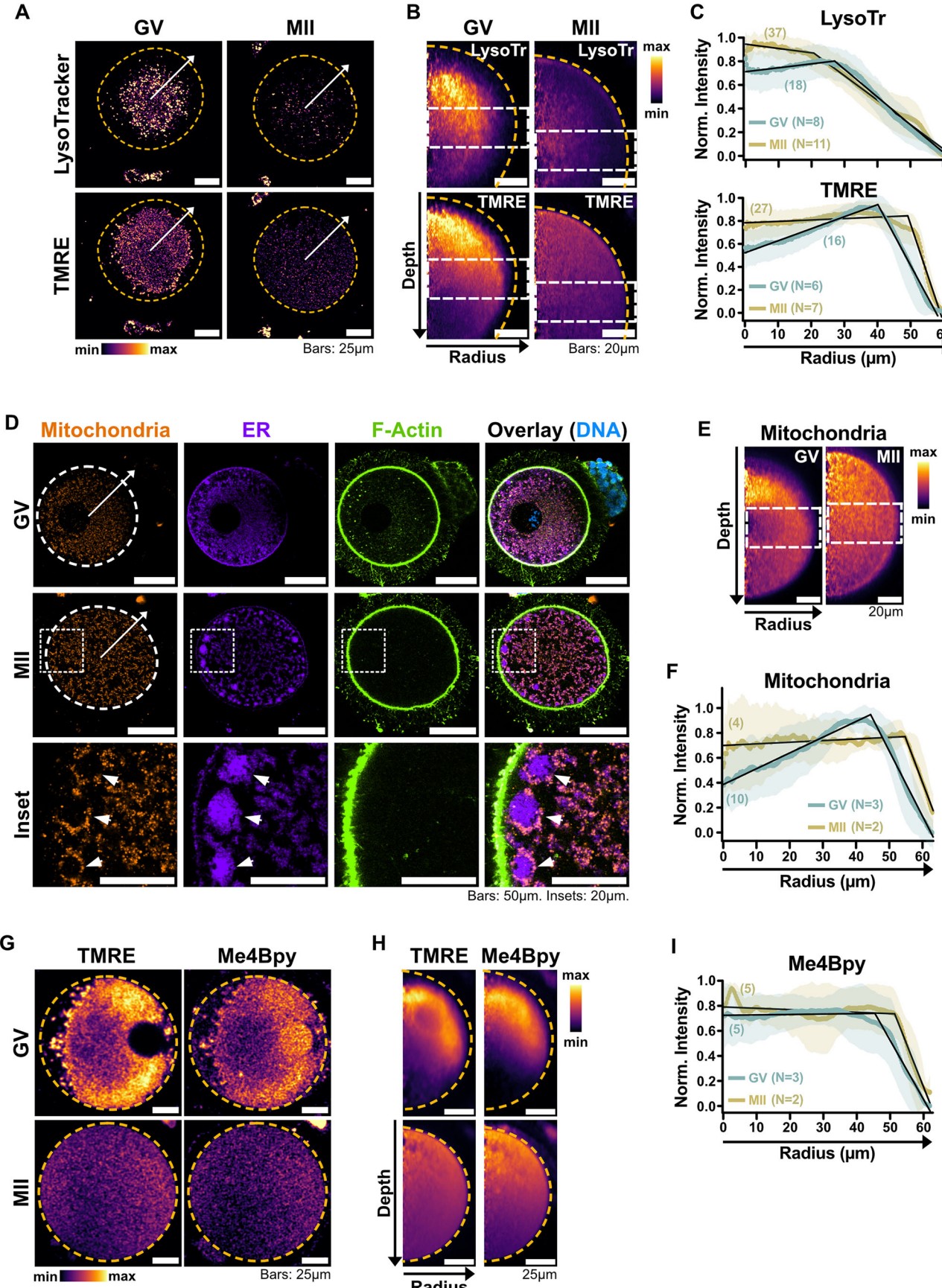

**Figure 4. Organelles are asymmetrically distributed in human oocytes.**

(A) Representative equatorial confocal planes of GV and MII oocytes labelled with LysoTracker and TMRE. Dashed circles indicate the oocyte boundaries. Arrows indicate the direction of the radius used for the radial reslicing shown in (B, C). (B) Radial reslicing of the oocytes shown in (A) along the depicted radiuses. Average projections of 360 radiuses spaced by 1° are shown. Orange dashed lines indicate the oocyte boundaries. White dashed boxes indicate the equatorial regions quantified in (C). (C) Radial distribution of LysoTracker (top) and TMRE (bottom) intensity in GV and MII oocytes, quantified as shown in (A, B). (D) Representative confocal images of GV and MII oocytes immunolabelled with anti-Calnexin (CANX) and anti-ATP5A to highlight the endoplasmic reticulum (ER) and mitochondria, respectively. The actin cytoskeleton was labelled with Phalloidin. Arrows indicate the direction of radiuses used for radial quantification shown in (F). Arrowheads indicate subcortical clusters of ER membranes surrounded by mitochondria in MIIs. (E) Radial reslicing of the oocytes shown in (D) along the depicted radiuses. Average projections of 360 radiuses spaced by 1° are shown. Dashed boxes indicate the equatorial regions quantified in (F). (F) Radial distribution of immunolabelled mitochondria in fixed oocytes, quantified as shown in (D, E). (G) Live confocal images of oocytes labelled with TMRE and Me4Bpy after Verapamil treatment. Orange dashed circles indicate the oocyte boundaries. See Fig. EV5A for additional pictures of the same oocytes for Lysotracker staining. (H) Radial reslice of the oocytes shown in (G). Orange dashed lines indicate the oocyte boundaries. (I) Radial distribution of Me4Bpy intensity in GV and MII oocytes. For (C, F), (I), numbers in parentheses indicate the total number of oocytes quantified per condition from multiple donors. N indicates the number of donors considered per condition. Mean and SD are shown. Data were normalized to the maximal and minimal intensities of each curve as 1 and 0, respectively. Black lines represent segmented models, confidence intervals (CIs) for GVs and MIIs do not overlap across conditions: Breakpoints of each model are as follows: Lysotracker GVs (27,012 μm; 95% CI 26,402–27,623 μm), Lysotracker MIIs (20,682; 95% CI 19,773–21,590 μm), TMRE GVs (40,118 μm; 95% CI 39,870–40,366 μm), TMRE MII (49,154 μm; 95% CI 48,820–49,489 μm), Mitochondria GV (44,377 μm; 95% CI 44,141–44,613 μm), Mitochondria MII (54,723 μm; 95% CI 54,481–54,965 μm), Me4Bpy GV (45,391 μm; 95% CI: 45,042–45,740 μm), Me4Bpy MII (51,336 μm; 95% CI 50,866–51,807 μm). Source data are available online for this figure.

**Table 1. Summary of the differences in waste disposal systems and mitochondrial activity during oocyte maturation in mice and humans.**

| Feature (during maturation) | Mouse oocyte | Human oocyte |
|---|---|---|
| Lysosome activity | Increases (Zaffagnini et al, 2024) | Decreases (this study) |
| Proteasome activity | Increases (Zaffagnini et al, 2024) | Decreases (this study) |
| Mitochondrial membrane potential | Increases (AL-Zubaidi et al, 2019; Cheng et al, 2022) | Decreases (this study) |
| Aggregates location | ELVAs (Zaffagnini et al, 2024; Satouh et al, 2024; Harasimov et al, 2024) | Refractile bodies (lysosomal) (this study) |

Sundström et al, 1985), mitochondria were clustered around the nucleus in GVs, while they were found across the whole cytoplasm in MIIs (Fig. 4D–F). Notably, in most MIIs (n = 7/9 oocytes from 5 donors) mitochondria formed hollow assemblies in the subcortical region of the oocyte, which were filled by ER membranes (Fig. 4D, arrowheads). ER/mitochondria assemblies were not detected in GVs (n = 5 oocytes from 2 donors) (Fig. 4D).

We next assessed the spatial distribution of Me4Bpy in Verapamil-treated GVs and MIIs. In GVs, cytoplasmic Me4Bpy fluorescence closely resembled the distribution of mitochondria, leaving an empty space under the cortex, while in MIIs it appeared distributed throughout the cytoplasm (Fig. 4G–I, see also Figs. 1C, EV1F, and EV2D).

Therefore, we conclude that organelles rearrange during oocyte maturation. Initially, active lysosomes, proteasomes, and mitochondria are unevenly distributed in human GV oocytes, while during oocyte maturation proteasomes and mitochondria redistribute throughout the cytoplasm (Fig. 4A–C,G–I). Specifically, we found that mitochondria rearrange to form subcortical clusters with the ER (Fig. 4D). We also found that residual lysosomes after exocytosis remain concentrated in the inner cytoplasm of MIIs (Figs. 2D and EV5B–D).

Together, our data demonstrate that human oocytes exhibit lower degradative activity compared to their neighbouring somatic cumulus cells. Additionally, we find that oocyte maturation involves the downregulation of lysosomal, proteasomal, and mitochondrial activities. Organelle clustering is also regulated by different means for different organelles in human oocytes: all three organelles we examined remain clustered close to the nucleus in GVs, but lysosomes stay clustered while mitochondria and proteasomes are redistributed in MIIs. Using over 100 human oocytes donated by healthy donors—most of which were donated

MIIs—our study marks a significant milestone in understanding organelle activity and distribution in healthy human oocytes.

## Discussion

Our study represents a landmark characterization of organelle activity and distribution in freshly donated human oocytes, under conditions closely aligned with natural biology and IVF applications. This is particularly significant because the reproduction field heavily relies on in vitro matured MIIs for basic research. However, in vitro matured MIIs fail to replicate the physiological conditions of natural maturation and are associated with poor clinical outcomes (Das and Son, 2023). By performing the first large-scale live characterization of organelle activity and distribution in freshly-donated healthy human oocytes, we uncovered several unique aspects of proteostasis regulation in these cells, compared to other species and cell types.

We hypothesize that the low organelle activity in human oocytes reflects a lower metabolic rate, possibly to prevent the accumulation of damaged intracellular components (Rodríguez-Nuevo et al, 2022). Intriguingly, this links our findings to the emerging field of developmental tempo regulation, which has identified slower protein degradation as a hallmark of human development compared to faster-developing species (Rayon et al, 2020; Nakanoh et al, 2024; Matsuda et al, 2024). Our results extend this concept by showing that reduced protein turnover is already established at the oocyte stage in humans and may persist into embryogenesis.

Our data reveal key differences in proteostasis regulation between human and mouse oocytes (Table 1), which we speculate can be explained by their different growth rates: mouse oocyte growth spans few weeks (Pedersen, 1970; Hoage and Cameron, 1976), while human oocytes develop over several months

(Gougeon, 1986). First, mouse oocytes store aggregated proteins in ELVAs (Zaffagnini et al, 2024), whereas human oocytes house aggregates in a few enlarged lysosomes (Fig. 3). The faster growth of mouse oocytes may result in the production of more protein aggregates that need to be managed differently. Second, while protein degradation and mitochondrial membrane potential increase during oocyte maturation in mice (Zaffagnini et al, 2024; AL-Zubaidi et al, 2019), both are instead downregulated in human oocytes (Fig. 1A–D). The downregulation of organelle activity in human MIIs could be part of a strategy to minimize reactive oxygen species production (Rodríguez-Nuevo et al, 2022) during the longer maturation process in humans. Third, we identified lysosomal exocytosis in human oocytes (Figs. 2D,F,G, EV3B,E,F, and EV5B–D), a process conserved in mice, though regulated differently between species. In mice lysosomal exocytosis peaks after fertilization and is coupled to ELVA disappearance (Zaffagnini et al, 2024), whereas in humans it occurs already during oocyte maturation (Figs. 2D,F,G, EV3B,E,F, and EV5B–D). The mechanisms and significance of lysosomal exocytosis during mammalian oocyte and embryo development remain to be elucidated.

Finally, our data show that organelles undergo a substantial redistribution during human oocyte maturation. We report large (10 µm) cortical ER-mitochondria clusters in most MIIs, whose function is unknown. Further studies will reveal whether the smooth endoplasmic reticulum aggregates that are associated with poor fertility outcomes in humans (Sá et al, 2011; Tatíčková et al, 2023) are a dysregulated form of these physiologically relevant ER-mitochondria clusters. Moreover, we found that lysosomes are regulated differently than mitochondria and most remain clustered in the centre of MII oocytes. These suggest that mitochondria and lysosomes are transported on different tracks in human oocytes.

Taken together, our findings overturn the assumption that increased proteostasis is a conserved feature of oocyte maturation. Instead, we reveal a distinct, human-specific programme characterized by reduced degradative and mitochondrial activity. These adaptations likely serve to minimize cellular stress over the prolonged timeline of human oocyte development and represent a new layer of species-specific reproductive strategy.

# Methods

### Reagents and tools table

| Reagent/resource | Reference or source | Identifier or catalogue number |
|---|---|---|
| **Experimental models** | | |
| Human oocytes, freshly isolated | n/a | n/a |
| **Recombinant DNA** | | |
| n/a | n/a | n/a |
| **Antibodies** | | |
| Ms anti-LAMP1 (H4A3) | Abcam | ab25630 RRID: AB_470708 |
| Rb anti-LAMP1 | SigmaAldrich | L1418 RRID: AB_477157 |
| Rb anti-CANX | Abcam | ab22595 RRID: AB_2069006 |
| Ms anti-ATP5A | Abcam | ab14748 RRID: AB_301447 |
| Rb anti-CTSD | Abcam | ab75852 RRID: AB_1523267 |
| Rb anti-20S | Abcam | ab22673 RRID: AB_2268907 |
| AlexaFluor-647 Gt anti-Mouse | Thermo Fisher | A-21236 RRID: AB_2535805 |
| AlexaFluor-555 Gt anti-Rabbit | Thermo Fisher | A-21428 RRID: AB_141784 |
| **Cell culture media** | | |
| G-Gamete | VitroLife | 10126 |
| G-IVF PLUS | VitroLife | 10134 |
| Ovoil | VitroLife | 10029 |
| **Dyes and inhibitors** | | |
| LysoTracker Deep Red | Thermo Fisher | L12492 |
| TMRE | Thermo Fisher | T669 |
| Me4Bpy | Bio-Techne | I-190-050 |
| Verapamil | Spirochrome | n/a |
| Bafilomycin A1 | Abcam | ab120497 |
| MG-132 | Merck Millipore | 474790 |
| CCCP | Abcam | ab141229 |
| Proteostat | Enzo | ENZ-51035-K100 |
| **Software** | | |
| FIJI | ImageJ.net | 2.14.04/1.54f |
| Prism 10 | GraphPad | 10.4.2 |

# Methods and protocols

### Limitations of the study

A few limitations of this study should be acknowledged. While this study includes, to our knowledge, the largest number of human oocytes analysed for organellar activity to date, the sample size is necessarily constrained by the limited availability and ethical considerations surrounding the use of healthy human oocytes. However, the relatively large number of oocytes in our study enabled paired analyses of sibling oocytes, strengthening the internal validity of our comparisons. That said, including more oocytes per donor or increasing the number of donors could provide greater statistical power to support the conclusions of this manuscript. In addition, our assessment of organellar function—specifically mitochondrial membrane potential, lysosomal and proteasomal activity—relied on live-cell fluorescent probes. Although these are widely accepted and commonly used tools in the field, they do not provide mechanistic insights. Future studies incorporating complementary functional assays that would involve multi-centre studies to increase oocyte numbers will be essential to further elucidate the molecular mechanisms underlying organelle regulation during oocyte maturation. Despite these limitations, our findings provide a robust and novel resource for understanding subcellular changes in human oocytes during maturation.

### Ethical approval

This study was approved by the ethics committee of Grupo Quironsalud Catalunya (Resolution 26/2023 of 19/12/2023) and by the ethics committee for clinical research of Parc de Salut Mar, Hospital del Mar Research Institute, Barcelona (project number: 2023/11385/I). The oocytes were donated for research according to Law 14/2006 on Assisted Human Reproduction. All donors received thorough information about the project and signed an informed consent form. Strict confidentiality of all personal and research data was ensured according to the current legislation (European Regulation 2016/679 of April 27, on the protection of natural persons with regard to the processing of personal data and on the free movement of such data (GDPR), and the Organic Law 3/2018 of December 5, on Personal Data Protection and guarantee of digital rights (LOPDGDD)). Samples and donors' information were transferred from Dexeus Mujer to CRG in anonymized form. The donors' information collected and included in the study consisted of donors' age, body mass index (BMI), and whether they have proven fertility, defined as at least one previous pregnancy after natural conception (Table EV1). This project complies with the guidelines of Law 14/2007 on Biomedical Research and with the Declaration of Helsinki and its latest version (Fortaleza, Brazil 2013).

### Donor recruitment and oocyte retrieval

All the recruited donors were between 18 and 35 years old, and fulfilled the clinical and legal requirements established by the Spanish legislation (Real Decreto-Ley 412/1996 and Real Decreto-Ley 9/2014). Anonymized donors' information is summarized in Table EV1. The oocyte donors underwent a rigorous screening for their ovarian reserve and good health status as described (Clua et al, 2010). Donors were stimulated and ovulation was triggered as previously described (Solé et al, 2013). Oocytes were retrieved 36 h after ovulation induction via ultrasound-guided transvaginal follicular aspiration. Following recommended oocyte collection procedures (D'Angelo et al, 2019), all safely accessible follicles larger than 13 mm were punctured, which are known to yield oocytes from GV to MII stage (McCulloh et al, 2020). All the GVs obtained were included in this study. Supernumerary MIIs assigned for research were selected randomly and were morphologically indistinguishable from the siblings used for IVF.

### Oocyte manipulation

After retrieval, COCs were cultured at 37 °C, 6% $CO_2$ 5% $O_2$ in G-IVF Plus™ (Vitrolife, Sweden) covered with Ovoil™ (Vitrolife) until transport to the lab. COCs were transported in G-Gamete™ (Vitrolife) at 37 °C and immediately processed upon reception. COCs were processed within 1.5–2 h from retrieval. COCs were dispersed in 500 μl G-Gamete supplemented with 1× Hyase (Vitrolife) under Ovoil at 37 °C, and pipetted with a 140 μm-bore Stripper (CooperSurgical) to dissociate cumulus cells. Oocytes were partially denuded and transferred to clean G-IVF at 37 °C using a 275 μm-bore Stripper and further processed for live imaging or directly fixed for immunofluorescence.

### Live imaging

LysoTracker was used at 50 nM, TMRE at 100 nM and Me4Bpy at 1 μM. Oocytes were loaded with dyes in G-IVF for 30-45 min at 37 °C 5% $CO_2$, washed in clean G-IVF and transferred for imaging

to a glass-bottom 35 mm dish (MatTek) in a small droplet of G-IVF covered with Ovoil. Oocytes were imaged at 37 °C 5% $CO_2$ with a Leica Stellaris 5 confocal microscope equipped with a 40x water-immersion objective or a 63x glycerol-immersion objective. Z stacks encompassing the whole oocyte volumes were taken with 1–2.5 μm spacing between slices. For Verapamil treatment, oocytes were loaded with dyes and imaged as described, then reloaded in the same medium supplemented with 10 μM Verapamil, washed and imaged again with the same settings. For BafA1/MG-132 treatment, oocytes were pre-incubated with 250 nM BafA1 + 10 μM MG-132 in G-IVF for at least 45 min, then loaded with dyes in presence or absence of BafA1 and MG-132 and imaged as described. For CCCP treatment, TMRE-laden oocytes were imaged with a Leica DMi8 epifluorescence microscope equipped with a ×40 air objective, then incubated with 50 μM CCCP in G-IVF medium for 15 min at 37 °C 5% $CO_2$, and imaged again with the same settings.

### Immunofluorescence

Oocytes were fixed in 2% formaldehyde (FA) in phosphate-buffered saline (PBS) 0.01% Poly-vinyl alcohol (PVA) for 1 h room temperature (RT), permeabilized in PBS 0.05% Triton X-100 for 30 min RT, blocked in PBS 3% bovine serum albumin 1% normal goat serum 0.1% Tween-20 for 1 h RT and incubated o/n at 4 °C in blocking buffer with primary antibodies. Oocytes were washed for 1 hr in blocking buffer at RT, incubated for 2 h in blocking buffer at RT with secondary antibodies, washed for 1 h in blocking buffer, rinsed in PBS PVA and spotted for imaging in PBS PVA under mineral oil. Imaging was performed with Leica SP5 or SP8 inverted microscopes equipped with a ×40 water-immersion lens.

For LAMP1 immunofluorescence on non-permeabilized oocytes, oocytes were fixed as described then directly incubated with Ms anti-LAMP1 H4A3 in blocking buffer without detergent for 1 h RT, washed for 30 min in PBS PVA, incubated with the secondary antibody for 30 min RT in blocking buffer without detergent, washed for 30 min RT in PBS PVA and spotted for imaging as described. All antibodies were used at 1:100 dilution, except Ms anti-LAMP1 H4A3, which was used at 1:25.

### Proteostat labelling

Oocytes were fixed in 2% FA in 1× Assay Buffer 0.01% PVA for 1 h RT, permeabilized as per the manufacturer's instructions in 1× Assay Buffer, 0.5% Triton X-100, 3 mM EDTA, pH 8, for 1 h on ice, then incubated o/n in PBS 0.01% PVA with Rb anti-LAMP1 (Sigma). Oocytes were washed in PBS PVA for 1 h RT, incubated in 1× Assay Buffer 0.01% PVA with the secondary antibody +/−Proteostat 1:2000 for 2 h RT, washed in PBS PVA for 1 h RT and spotted for imaging as described.

### Image quantification

All images were analysed with FIJI. The resulting data were plotted and statistically compared with GraphPad Prism 10.

For the quantification of the oocyte diameter, the major axis of the best fitting ellipse at the equatorial plane of the oocyte was taken as readout. Only oocytes imaged live were used for the quantification of the diameter, to prevent potential artefacts of fixation.

The assessment of the chromatin configuration in GVs was performed on oocytes incubated with a DNA dye (Hoechst 33342) according to the classification by Combelles et al (2002). For

LysoTracker, TMRE, Me4Bpy, Proteostat and 20S comparison between GVs and sibling MIIs, a maximal *Z* projection of the whole stack for each oocyte was made, then the mean intensity of the whole oocyte area was taken as readout. For the quantification of the mean intracellular LAMP1 intensity, a maximal projection across the equatorial region of each oocyte was made, and the mean intensity of the largest possible cytoplasmic area (excluding the PM signal) was taken as readout. Only donors with at least one GV and one MII were taken into account.

For LysoTracker, TMRE, Me4Bpy and 20S comparison between oocytes and cumulus cells, both cell types were segmented based on their size, then the average intensity of the segmented areas in each slice was calculated. Intensities were averaged across stacks and plotted. For each stack, only slices with both the oocyte and cumulus cells were taken into account. To avoid artefacts of potential dead cells, only cumulus cells with positive TMRE labelling were considered for the analysis. Similarly, for 20S quantification, only cumulus cells with both nuclear and cytoplasmic 20S signal were considered for analysis. To account for potential signal dilution due to the large size of oocytes, we implemented several control experiments. Fixed oocytes and surrounding somatic cells stained for organelle markers (e.g. LAMP1, ATP5A) showed comparable immunofluorescence intensity, suggesting similar organelle content. In contrast, live-cell labelling with activity probes (e.g. LysoTracker, TMRE) revealed lower intensity in oocytes, indicating functional differences. To control for dye efflux, we used verapamil to inhibit ABC transporters and found that somatic cells still retained higher probe intensity. Finally, we performed LysoTracker staining on mouse oocytes and cumulus cells, where signal intensity was comparable—consistent with known high lysosomal activity in mouse oocytes—supporting that differences observed in human samples reflect biological specificity rather than size-related artefacts.

For the quantification of the radial distribution of LysoTracker, TMRE, Me4Bpy and immunolabelled mitochondria, all the quantified oocytes were first rescaled to have the same pixel size. Then each oocyte centre was defined as the centroid of an ellipse fitting the oocyte shape at the equator. A radius was drawn from the centre to the oocyte edge and the stack was radially resliced across 360° with 1° increments. Radial reslices were average-projected and a box was drawn encompassing the oocyte equatorial region, then the average intensity within the box was plotted along the radius and taken as readout. Each curve was normalized by setting its minimum and maximum as 0 and 1, respectively, then curves were averaged among oocytes.

For LAMP1 PM quantification, oocytes were radially resliced and the average intensity in the equatorial region was plotted as described. The maximum intensity of the PM peak was taken as readout for each oocyte.

For the calculation of the breakpoints, an R script was used to calculate the mean and standard deviation of the data at each distance along the projected radius. A basic linear model was then fitted with the lm() function, which establishes an initial relationship between the mean measurement and distance. Then, the segmented package was used to perform a piecewise linear regression by applying the segmented() function to the initial model. This function iteratively identifies a breakpoint in the data and adjusts the model to fit distinct linear segments on either side of that breakpoint. Breakpoints positions for each fit, with their confidence intervals, were then calculated and compared.

## Data availability

All data supporting the findings of this study are available within the article, extended view data, and the associated data tables. ImageJ macros and the R script used for image segmentation and graph analysis are available on Zenodo (https://doi.org/10.5281/zenodo.15109513).

The source data of this paper are collected in the following database record: biostudies:S-SCDT-10_1038-S44318-025-00493-2.

## Peer review information

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

## Acknowledgements

We are grateful to the CRG Advanced Light Microscopy Units for the support and assistance in this work. E.B. acknowledges institutional core funding and the support of the Spanish Ministry of Science and Innovation through the Centro de Excelencia Severo Ochoa (CEX2020-001049-S, MCIN/AEI/10.13039/501100011033) and the Generalitat de Catalunya through the CERCA programme, the Ministerio de Ciencia e Innovación PID2023-146697NB-I00 funded by MCIN/AEI/10.13039/501100011033/FEDER, UE and the Vallee Scholar Award (Vallee Foundation Inc).

## Author contributions

**Gabriele Zaffagnini**: Data curation; Software; Formal analysis; Validation; Investigation; Visualization; Methodology; Writing—original draft; Project administration; Writing—review and editing. **Miquel Solé**: Resources; Data curation; Investigation; Methodology; Project administration; Writing—review and editing. **Juan Manuel Duran**: Data curation; Formal analysis; Investigation; Visualization; Methodology; Writing—review and editing. **Nikolaos P Polyzos**: Resources; Supervision; Methodology; Writing—review and editing.

**Elvan Böke**: Conceptualization; Resources; Supervision; Funding acquisition; Writing—original draft; Project administration; Writing—review and editing.

Source data underlying figure panels in this paper may have individual authorship assigned. Where available, figure panel/source data authorship is listed in the following database record: biostudies:S-SCDT-10_1038-S44318-025-00493-2.

## Disclosure and competing interests statement

The authors declare no competing interests.

# Expanded View Figures

**Figure EV1. Morphometric analysis of the retrieved oocytes and validation of LysoTracker, TMRE and Me4Bpy labelling in human cumulus-oocyte complexes.** ▶

(**A**) Quantification of the oocyte diameter from live-cell imaging experiments. Data points represent individual oocytes and are colour-coded by donor (see Table EV1). The horizontal lines indicate the median of each distribution, boxes represent the interquartile ranges (IQR), and whiskers denote the minima and maxima, respectively. Numbers in parentheses indicate the total amount of oocytes quantified per condition from $N = 9$ and $N = 12$ donors, respectively. $p$ value: unpaired $t$ test with Welch's correction. (**B**) Representative confocal images of the chromatin configurations in GVs isolated from $N = 7$ donors and labelled with Hoechst 33342. Classification was performed according to Combelles et al (2002). The number of oocytes in each class is indicated. (**C**) Relative proportion of the chromatin configuration classes among the analysed GVs. The original data from Combelles et al (2002) are reported for comparison. Number into parentheses indicate the total amount of oocytes scored per study. The $p$ value was calculated with a Fisher's exact test on the raw oocyte counts. (**D**) Representative live confocal images of Zona pellucida-attached cumulus cells unlabelled or labelled with LysoTracker Deep Red, Me4Bpy and TMRE to highlight active lysosomes, proteasomes, and mitochondria, respectively. (**E**) Representative live epifluorescence images of the same oocyte before and after CCCP treatment to dissipate the mitochondrial membrane potential. Mitochondria were labelled with the membrane potential-sensitive dye TMRE before CCCP treatment. (**F**) Live confocal images of GV-stage oocytes treated with or without Bafilomycin A1 and MG-132 to inhibit lysosomes and proteasomes, respectively. Oocytes were labelled in presence of Verapamil with LysoTracker, Me4Bpy and TMRE to highlight active lysosomes, proteasomes and mitochondria, respectively. DNA was counterstained with Hoechst 33342. (**G**) Representative live confocal images of Zona-attached cumulus cells treated with or without Bafilomycin A1 and MG-132 to inhibit lysosomes and proteasomes, respectively. Cells were labelled with LysoTracker Deep Red, Me4Bpy and TMRE to highlight active lysosomes, proteasomes and mitochondria, respectively. Notice that treated cells were still alive, as indicated by the retention of TMRE signal. Source data are available online for this figure.

**A**

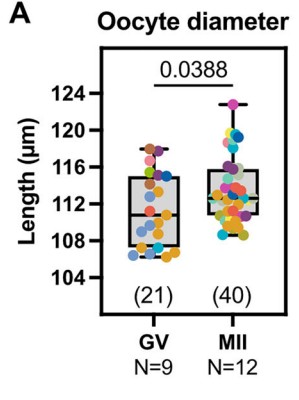

Oocyte diameter

**B** Chromatin configuration (GVs)

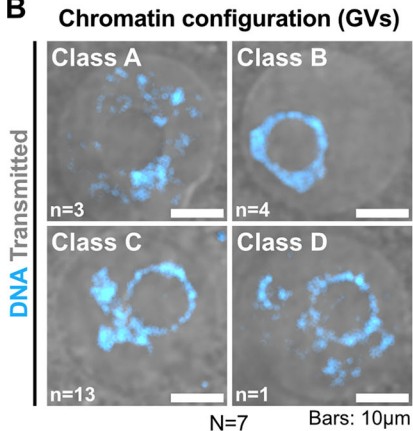

**C** Chromatin configuration (GVs)

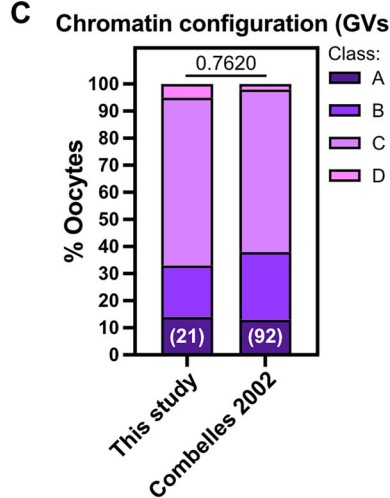

**D**

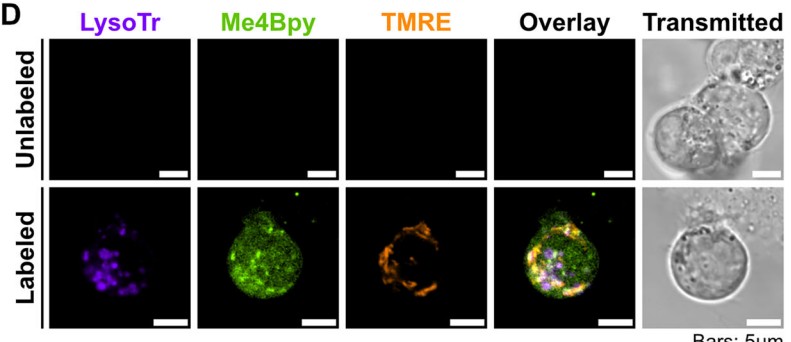

**E**

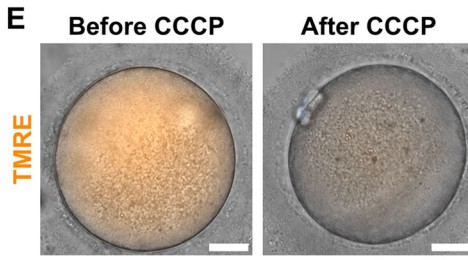

Bars: 25μm

**F**

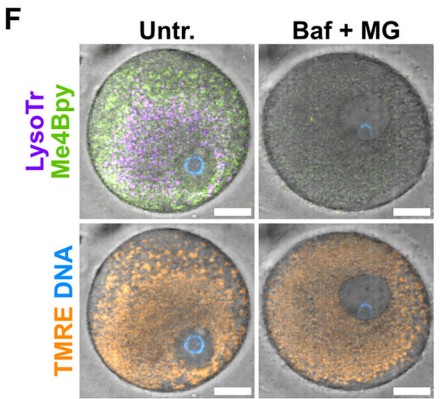

Bars: 25μm

**G**

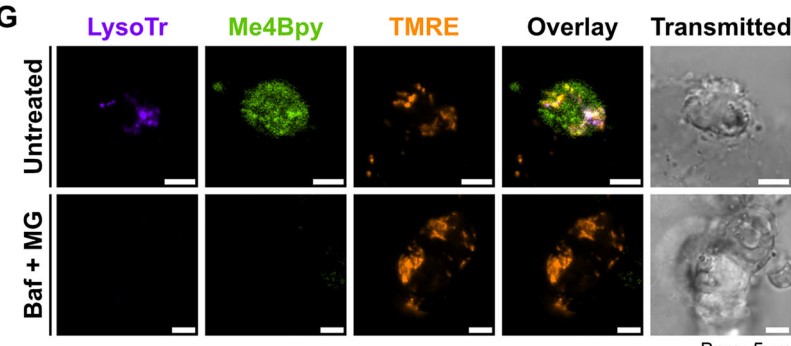

Bars: 5μm

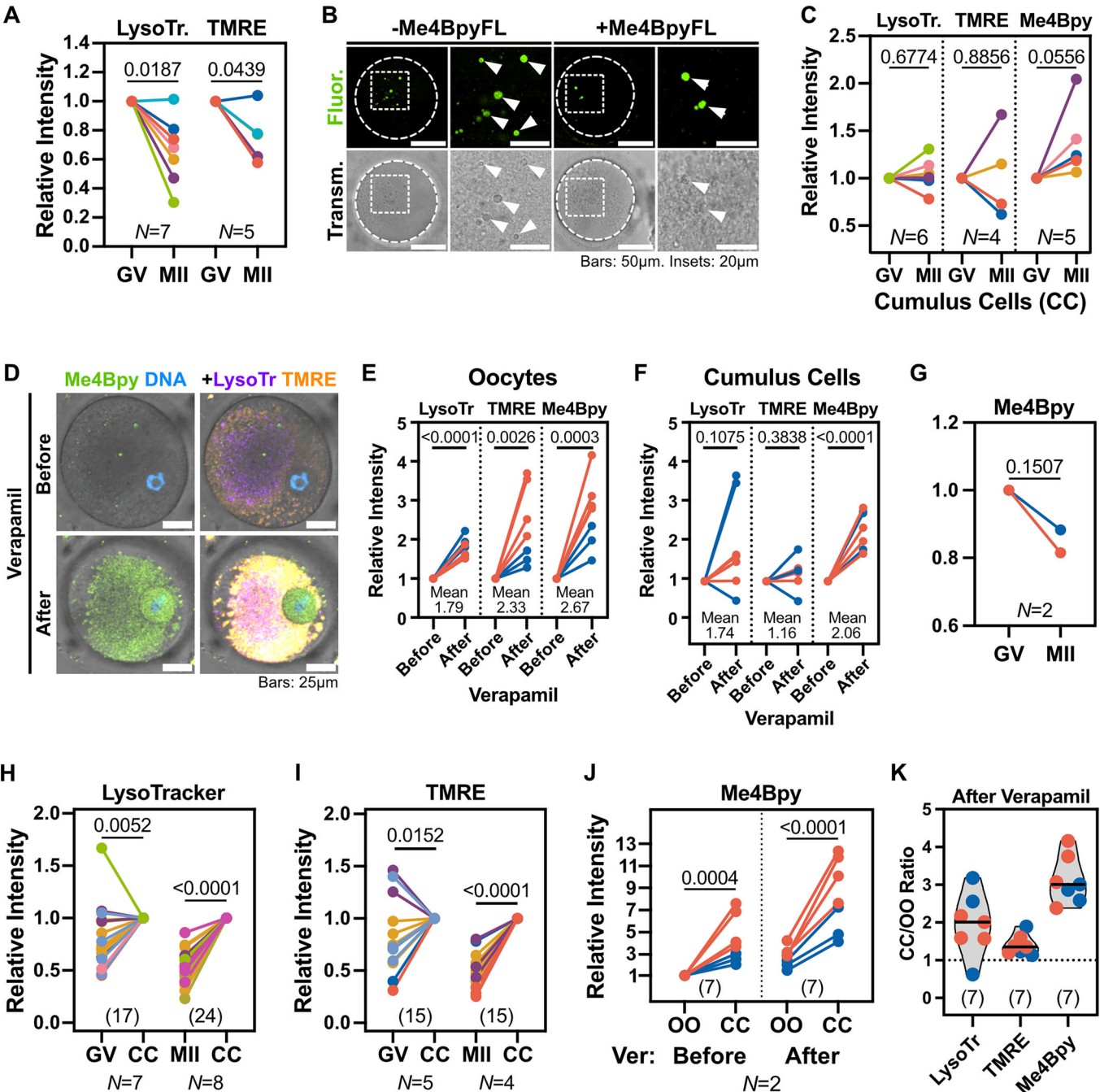

◄ **Figure EV2. Characterization of LysoTracker, TMRE, and Me4Bpy in oocytes and cumulus cells before and after Verapamil treatment.**

(A) Comparison of the average Lysotracker (lysosomes) and TMRE (mitochondria) intensity in GV and MII oocytes as shown in Fig. 1B. Data from multiple oocytes were averaged by donor and normalized to the GV value for each donor. $N$ numbers indicate the total donors considered for each quantification. $p$ values: ratio-paired $t$ tests. (B) Representative live confocal images of oocytes labelled with or without Me4Bpy. Arrowheads indicate autofluorescent refractile bodies. (C) Comparison of LysoTracker, TMRE and Me4Bpy intensity in cumulus cells attached to GV- and MII-stage oocytes from multiple donors. Data were averaged by donor and normalized, for each donor and dye, to the GV value. Only donors with at least one GV and one MII were considered. $p$ values: ratio-paired $t$ tests. (D) Representative live confocal images of the same oocytes labelled with LysoTracker, TMRE and Me4Bpy and imaged before and after incubation with Verapamil. DNA was counterstained with Hoechst 33342. (E, F) Quantification of LysoTracker, TMRE and Me4Bpy intensities in the same oocytes (E) and cumulus cells (F) before and after verapamil addition. Data points represent individual COCs from $N = 2$ donors. For each COC, data were normalized to the value before verapamil. $p$ values: ratio-paired $t$ tests. (G) Quantification of the average Me4Bpy intensity in Verapamil-treated GV and MII oocytes from $N = 2$ donors, as shown in Fig. 1D. Data from multiple oocytes were averaged by donor and normalized to the GV value for each donor. $p$ value: ratio-paired $t$ test. (H, I) Pairwise comparison of Lysotracker (H) and TMRE (I) intensities between oocytes and the corresponding cumulus cells (CC) as shown in Fig. 1E,F. Data points represent individual Cumulus Oocyte Complexes (COCs) and are colour-coded by donor. For each COC, data were normalized to the CC value. $p$ values: ratio-paired $t$ tests. (J) Pairwise quantification of Me4Bpy intensity in the same oocytes (OO) and cumulus cells (CC) before and after incubation with Verapamil (Ver) as shown in Fig. 1G. Data points represent individual COCs and were normalized to the OO value before Verapamil for each COC. $p$ values: ratio-paired $t$ tests. The comparison between oocyte intensities before and after Verapamil treatment is shown in Fig. EV2E. (K) Quantification of the LysoTracker, TMRE and Me4Bpy intensity ratio between each oocyte (OO) and its corresponding cumulus cells (CC) after verapamil addition.

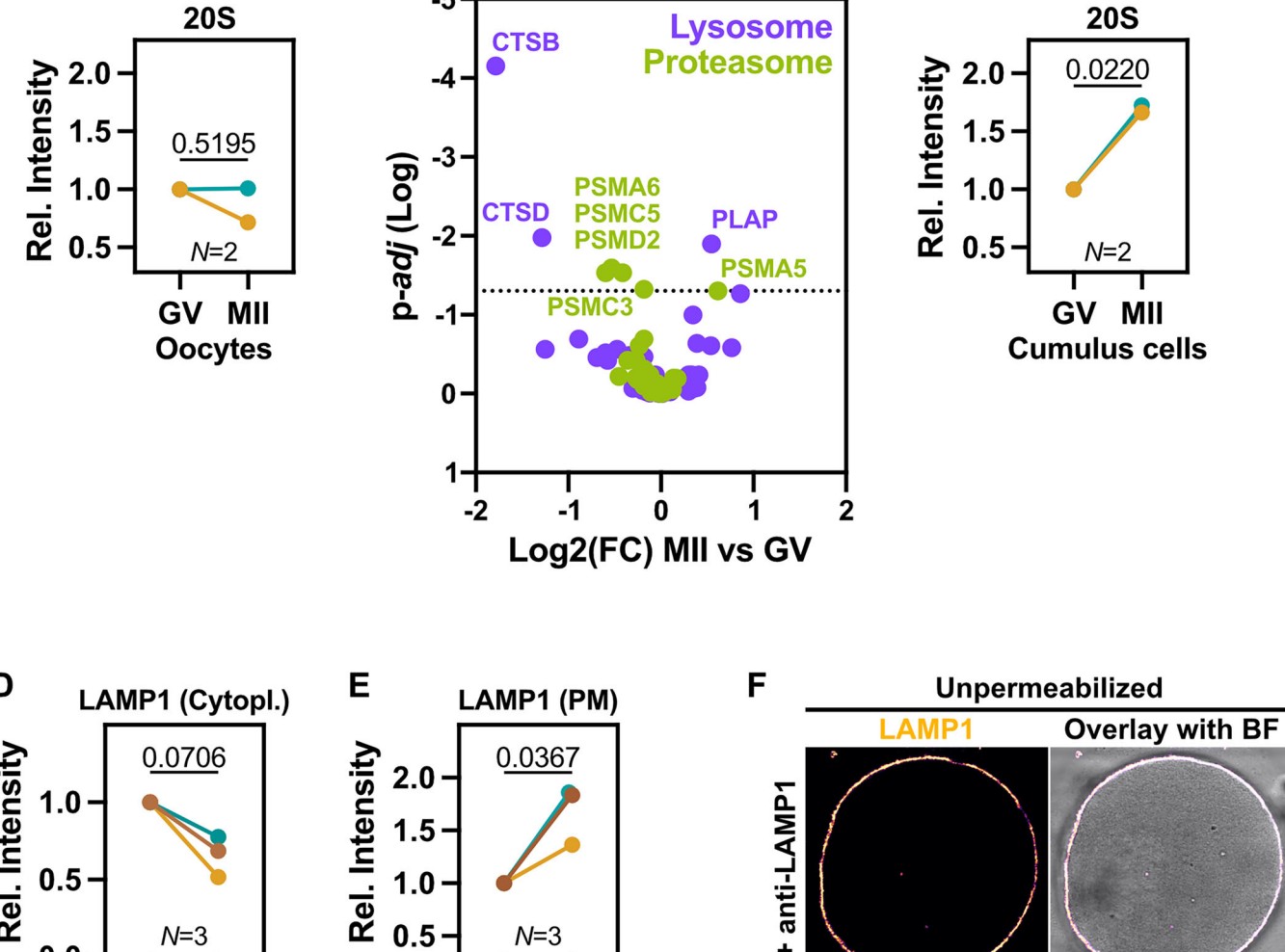

**Figure EV3.  The reduction in lysosome levels in MIIs is accompanied by lysosomal exocytosis.**

(A, C) Comparison of the average 20S intensity in immunolabelled oocytes (A) and cumulus cells (C) from $N = 2$ donors. Data from multiple oocytes were averaged by donor and normalized to the GV value for each donor. *p* values: ratio-paired *t* tests. (B) Relative change in lysosomal (indigo) and proteasomal (green) proteins between GVs and MIIs, as detected by single-cell proteomics of human oocytes. Data were retrieved from (Galatidou et al, 2024). Proteins were selected based on GO Cellular Component (CC) terms matching "lysosome" and "proteasome", respectively. The dashed line indicates $p = 0.05$. (D, E) Quantification of the mean intracellular (D) or plasma membrane (E) LAMP1 intensity in GV and MII oocytes shown in Fig. 2D–F. For each donor ($N = 3$), data from multiple oocytes were averaged and normalized to the GV value. *p* values: ratio-paired *t* tests. (F) Representative confocal images of non-permeabilized MII oocytes labelled with or without anti-LAMP1 H3A4. LAMP1 intensity is displayed as a heatmap.

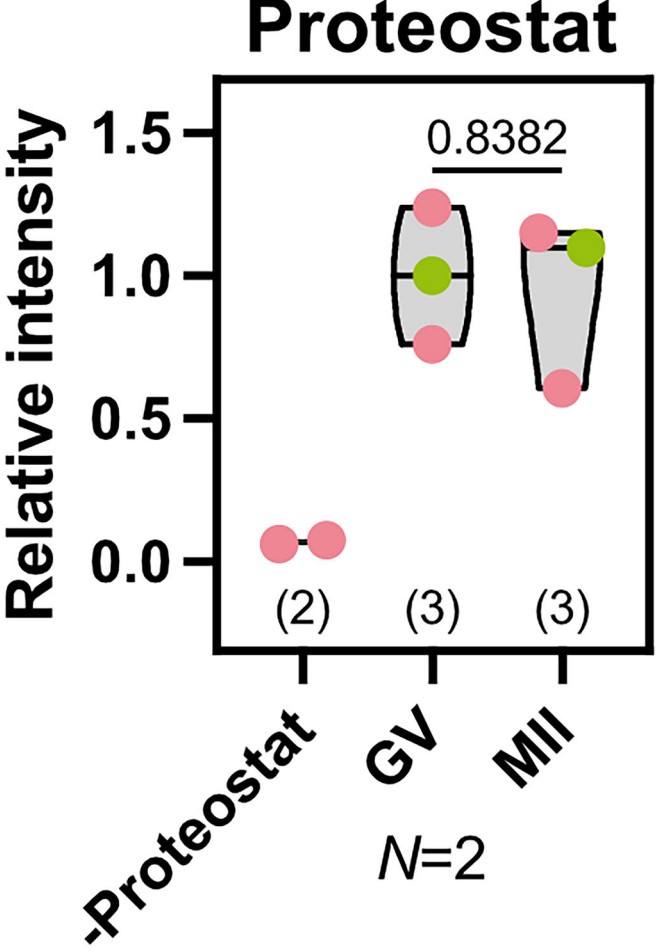

**Figure EV4.   Proteostat intensity does not change between GVs and MIIs.**

Quantification of mean proteostat intensity in refractile bodies in GV and MII oocytes from $N = 2$ donors. Only donors with at least 1 GV and 1 MII were considered. Data points represent individual oocytes and were normalized to the median of GVs for each donor. Numbers in parentheses indicate the number of oocytes quantified per condition. *p* value: unpaired *t* test with Welch's correction.

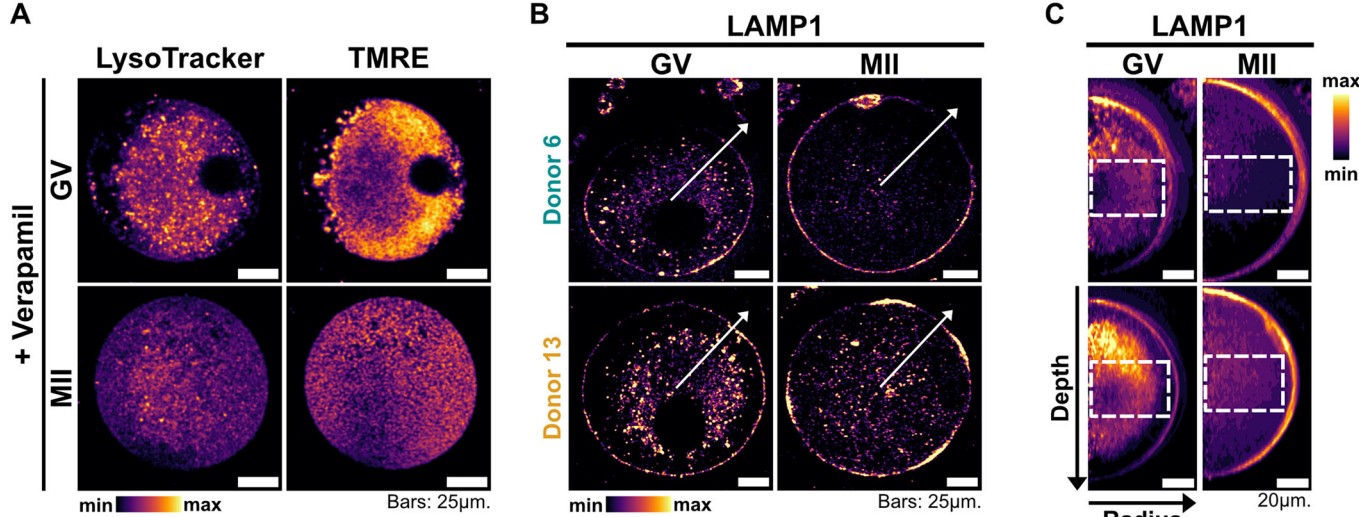

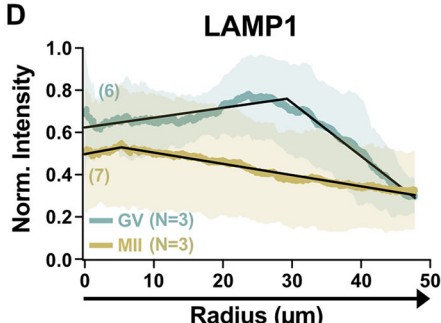

**Figure EV5.   Lysosomes are clustered in both human GVs and MIIs.**

(**A**) Live confocal images of GV and MII oocytes labelled with LysoTracker and TMRE after Verapamil incubation. Intensities are displayed as heatmaps. Additional images of the same oocytes are shown in Fig. 4G. (**B**) Confocal images of GV and MII oocytes from multiple donors immunolabelled with anti-LAMP1 (shown as heatmap). Arrows indicate radiuses used for the radial reslicing shown in (**C, D**). Examples of oocytes from a third donor are shown in Fig. 2D. (**C**) Radial reslicing of the oocytes shown in (**B**) along the depicted radiuses. Average projections of 360 radiuses spaced by 1° are shown. Dashed boxes indicate the equatorial regions quantified in (**D**). (**D**) Radial distribution of LAMP1 signal in the cytoplasm of fixed oocytes. Mean and SD are shown. Numbers in parentheses indicate the total amount of oocytes quantified per condition from multiple donors. *N* indicates the number of donors considered per condition. Data were normalized to the min and max of each curve as 0 and 1, respectively. Note that the lower levels of LAMP1 antibody staining on MIIs cause a low signal-to-noise ratio, increasing the deviation on its average curve. Radius length does not include plasma membrane in these measurements. Black lines represent segmented models. Breakpoints of each model are as follows: GVs (29,260; 95% CI 28,836–29,684 µm), MIIs (5308 µm; 95% CI 4573–6043 µm).

