## [Peer Review File · The EMBO Journal]

The proteostatic landscape of healthy human oocytes

Elvan Boke, Gabriele Zaffagnini, Miquel Solé, Juan Duran, and Nikolaos Polyzos

Corresponding author(s): Elvan Boke (elvan.boke@crg.eu)

Review Timeline:

Submission Date:	7th May 25
Editorial Decision:	28th May 25
Revision Received:	3rd Jun 25
Accepted:	6th Jun 25

Editor: Ieva Gailite

Transaction Report:

This manuscript was transferred to The EMBO JOURNAL following peer review at another journal.

Reviewers' comments:

We thank both reviewers for their excellent suggestions, which have undoubtedly strengthened our manuscript. In addition to addressing their individual comments, we have added a new table to the discussion to make clear the differences between mouse and human oocytes (Table 1, pasted below) and a paragraph to the discussion to contextualize our study within the broader literature, particularly in relation to recent advances on developmental tempo differences between species and their correlation with protein degradation, which we paste below.

“We hypothesize that the low organelle activity in human oocytes reflects a lower metabolic rate, possibly to prevent the accumulation of damaged intracellular components [1]. Intriguingly, this links our findings to the emerging field of developmental tempo regulation, which has identified slower protein degradation as a hallmark of human development compared to faster-developing species [2–4]. Our results extend this concept by showing that reduced protein turnover is already established at the oocyte stage in humans and may persist into embryogenesis”.

Feature (during maturation)	Mouse Oocyte	Human Oocyte
Lysosome activity	Increases [5]	Decreases (this study)
Proteasome activity	Increases [5]	Decreases (this study)
Mitochondrial membrane potential	Increases [6,7]	Decreases (this study)
Aggregates location	ELVAs [5,8,9]	Refractile bodies (lysosomal) (this study)

Table 1. Summary of differences in waste disposal systems and mitochondrial activity during oocyte in mouse and human.

All major changes in the main text are highlighted in **yellow** in the manuscript file.

Reviewer #1 (Remarks to the Author):

In this resource manuscript, the authors present analysis of the “proteostatic landscape” of human oocytes obtained from 21 healthy oocyte donors. The authors note that as much of our data from human oocyte studies often rely on oocyte donation from IVF patients, we are uncertain how reliable the findings are with respect to the fertile population. This may or may not be the case, as some studies use oocytes from male-factor indicated couples or gamete donors. Therefore, each study should be considered and not necessarily lumped together.

We thank the reviewer for their thoughtful feedback. We would like to clarify that our concern is not with the use of patient-derived oocytes per se, but rather with the use of in vitro maturation (IVM) to generate metaphase II (MII) oocytes from germinal vesicle (GV) oocytes. IVM-derived MIIs often fail to replicate the physiological conditions of natural maturation and are associated with poor fertility outcomes (reviewed in [10] and others). For this reason, we believe that

reference studies of oocyte metabolism and proteostasis should be conducted using in vivo-matured, freshly collected human oocytes.

However, most cellular biology studies to date have used “unsuitable-for-IVF/ICSI” GV oocytes that are then matured in vitro for experimental use (see [7,11–14], among others). To our knowledge, this manuscript is the most comprehensive cell biology study to date using freshly obtained in vivo-matured MIIs from healthy donors. The only comparable study examined just five such oocytes from five individuals [15]. That said, we agree with the reviewer that the distinction between IVM and in vivo MIIs should be clearly stated in the manuscript, and oocyte numbers should not be aggregated without context. Accordingly, we have added the following clarifications:

Abstract: “Using over 100 freshly donated oocytes from 21 healthy women aged 19–34.”

Introduction: “Using a substantial sample size—over 70 MIIs and 30 GVs collected directly from 21 healthy donors—”

“We did not use in vitro maturation (IVM) in this study, as IVM leads to suboptimal fertility outcomes [10].”

Regardless, what is unique here is the evaluation of two protein homeostasis pathways using simple staining and image methods to capture unique features of human oocytes and MII eggs. Here they examined ~30 GV oocytes and ~70 MII eggs and find that protein turnover is slow and declines, suggesting mechanisms to keep long-live proteins in the cell and a way to keep the cytoplasm full of proteins for the embryo. This study also highlights the need for also conducting studies on human samples as model organisms may have differences that should be noted in order to improve fertility practices. Therefore this study is quite original in the material they are using. I have a few questions and points for the authors to consider.

Thank you for the positive feedback!

1. The GVs used are from follicles that didn't respond to hormone stimulation. The MIIs did respond. Therefore, this implies that the GV oocytes were not meiotically competent, and immature compared to a fully grown oocyte ready to resume meiosis. Can the authors comment on this concern? Is the size of the GV oocytes or follicles fully grown or were they smaller? This knowledge becomes important because of the GV/MII comparisons and conclusions drawn.

We thank the reviewer for this question that gives us the chance to clarify our findings. Our goal was indeed to compare **in vivo-isolated** GVs and MIIs to assess potential differences in their proteostatic status. This investigation was motivated by our findings in mice, which show that mouse GVs are in a more repressed degradative state, whereas maturation triggers lysosomal acidification and proteasomal activation, and thus, leading to increased degradative activity in MIIs [5].

To achieve this, we specifically collected immature oocytes (GVs) and MIIs. This process followed the standard procedures at our collaborating hospital, Dexeus, where all safely

accessible follicles larger than 13 mm in diameter are punctured as part of their standard oocyte retrieval procedure to not leave any large follicles behind [16]. These follicles (>13mm) are known to yield both GV and MII oocytes [17]. As previously reported, GVs were typically obtained from smaller follicles [17]. We further find that, as expected [18], the retrieved GVs were on average slightly smaller than MIIs (Fig R1A). When we assessed the chromatin configuration of the obtained GVs (according to the widely accepted classification by Combelles et al. [19]), the results closely matching the previously reported findings (Fig R1B-C). The majority of oocytes (n=17/21) belonged to classes B and C, which are the most capable of resuming meiosis and completing maturation in vitro [20]. We have now included this information in the revised manuscript (Fig. S1A-C and methods) and clarified that our goal was to compare GVs and MIIs in humans. Interestingly, human GVs and MIIs display different (i.e opposite) proteostatic changes during maturation than do their counterparts in mice (Figure 1B, D vs Zaffagnini et al. [5], and the [new] Table 1).

Fig. R1: Morphometric analysis of retrieved GVs and MIIs. A) Quantification of the oocyte diameter from live-cell imaging experiments. Data points represent individual oocytes and are color-coded by donor (see Table S1). Numbers into parentheses indicate the total amount of oocytes quantified per condition from N=9 and N=12 donors, respectively. p-value: unpaired t-test with Welch’s correction. B) Representative confocal images of the chromatin configurations in GVs isolated from N=7 donors and labeled with Hoechst 33342. Classification was performed according to Combelles et al. [19]. The number of oocytes in each class is indicated. C) Relative proportion of the chromatin configuration classes among the analysed GVs. The original data from Combelles et al. [19] are reported for comparison. Numbers into parentheses indicate the total amount of oocytes scored per study. The p-value was calculated with a Fisher’s exact test on the raw oocyte counts.

2. The authors state that there was “no specific Me4Bpy accumulation in either oocyte stage.” What are the foci in the oocytes? It appears that there are several foci in MII oocytes compared

to GV. Therefore, what does significant accumulation look like? Perhaps quantify the number of foci would make this point stronger and more clear.

We thank the reviewer for highlighting this point. We find that the spots in the Me4Bpy channel in both GVs and MII oocytes correspond to autofluorescence of the refractile bodies, as shown in figure S2B of the revised manuscript, in which the same signal was also recorded in the absence of Me4Bpy (i.e, in the negative control) This was only noted in the figure legends in our first submission, but we have now made it clearer in the manuscript, and we apologize for the confusion.

3. The authors make intensity comparisons between oocytes which are large and allow more signal diffusion and cumulus cells which are small and would concentrate signal. Was this size difference taken into account? If so, how and can that be explained in the main text.

We thank the reviewer for this insightful comment. We agree that the large size of oocytes could, in principle, lead to signal dilution and complicate direct intensity comparisons with smaller somatic cells. To address this, we incorporated several strategies into our experimental design to distinguish true biological differences in organelle activity from potential size-related artefacts.

First, we performed immunofluorescence (IF) staining on fixed oocytes and surrounding somatic cells using organelle markers (e.g., LAMP1 and ATP5A). In fixed samples, oocyte organelles exhibited similar labeling intensity to those in somatic cells (Fig. R2A, B), suggesting that organelle content per se is not markedly different. This served as a baseline control for interpreting live-cell activity probe signals (e.g., LysoTracker, TMRE). In contrast to IF results, live staining with activity probes revealed lower intensity in individual oocyte lysosomes and mitochondria compared to somatic cell counterparts (Fig. R2A, B), indicating functional differences rather than technical dilution.

Second, we considered potential dye efflux as a confounding factor. Using verapamil to inhibit ABC transporters, we observed that somatic cells still retained higher probe intensity than oocytes (Fig. S2K of the revised manuscript), further supporting that the reduced labeling in oocytes reflects lower organelle activity rather than technical artifacts.

Finally, to experimentally address the concern about cell size, we performed the same LysoTracker staining on mouse oocytes adjacent to somatic cells. We already know that, in contrast to human oocytes, fully grown mouse oocytes have high lysosomal activity, comparable to somatic cells [5]. Consistently, LysoTracker intensity was similar between mouse oocytes and their adjacent cumulus cells (Fig. R2C), arguing that cell size alone does not account for signal differences. This further supports our conclusion that the reduced labeling observed in human oocytes is a biologically specific feature rather than an imaging artefact.

We have now clarified these methodological considerations and added supporting information to the revised manuscript in the methods section. Together, these results strongly suggest that the

observed differences in labeling reflect genuine differences in organelle activity between oocytes and somatic cells.

Fig. R2: Lower organelle activity labeling in oocytes is not due to more diluted signal. A) Left: representative images of human oocytes and cumulus cells labelled with TMRE (live) or anti-ATP5A (fixed) to highlight mitochondria. Single confocal planes are shown. Right: intensity plots of the TMRE/ATP5A signal along the indicated lines in oocytes (OO) and cumulus cells (CC). **B)** Left: representative images of human GV-stage oocytes and cumulus cells labelled with LysoTracker (live) or anti-LAMP1 (fixed) to highlight lysosomes. Single confocal planes are shown. Right: intensity plots of the LysoTracker/LAMP1 signal along the indicated lines in oocytes (OO) and cumulus cells (CC). **C)** Left: representative images of mouse GV-stage

oocyte and cumulus cells (Cumulus-oocyte-complex, COC) labelled with LysoTracker as in B. A single confocal section is shown. Right: intensity plot of the LysoTracker signal along the indicated lines in the oocyte (OO) or in cumulus cells (CC).

4. Did the authors see any differences in trends in the younger donors vs the older donors? Stratifying under/over 27y could be interesting.

We thank the reviewer for this excellent suggestion. We plotted the average LysoTracker and TMRE intensity of the oocytes shown in Fig. 1A-B and 1E-F against the respective donors' age. The data were normalised either to GVs (oocytes from Fig. 1A-B) or to cumulus cells (oocytes from Fig. 1E-F) for each donor. This analysis suggested a negative correlation between the donor's age and TMRE intensity in MII oocytes, – though not in GVs–, depending on the normalization performed (Fig. R3A-C). While there appears to be a trend toward lower TMRE intensity in older MII oocytes, we would rather not include this information in the manuscript due to the limited sample size. We instead plan to pursue a more targeted study to explore the effect of age on these parameters in the future.

Fig R3: TMRE intensity reduction from GV to MII is more pronounced with increasing donors' age. **A)** The average LysoTracker and TMRE intensities in MII oocytes were plotted against the age of the respective donor. Data were averaged by donor and normalized to the average of GVs for each donor as in Fig. 1B, then fitted to a line. Linear regression coefficients (R^2) are indicated for both datasets. Numbers indicate donors as in Table S1. **B-C)** Correlation between LysoTracker (A) and TMRE (B) intensity and donors' age. Data from individual oocytes, normalized to the respective CCs as in Fig. 1E-F, were averaged by donor and plotted against the corresponding donor's age, then fitted to a line. Linear regression coefficients (R^2) are indicated for both datasets. Numbers indicate donors as in Table S1.

5. I'm not fully following how the authors know that a human oocyte doesn't have an ELVA. Could a refractile body be an ELVA?

Endolysosomal vesicles assemblies (ELVAs) are clusters of several endo-lysosomal organelles such as endosomes, lysosomes and autophagosomes, held together by a proteinaceous glue [5]. In contrast, refractile bodies in human oocytes are enlarged vesicles (i.e no protein glue) surrounded by a single membrane of lysosomal origin ([21] and our own data Fig. 3A). Thus, the refractile bodies do not represent ELVAs embedded in a non-membrane-bound proteinaceous matrix [5].

Reviewer #2 (Remarks to the Author):

The extreme longevity of mammalian oocytes, mainly studies in mice, is associated with the presence of exceptionally long-lived proteins. In the present manuscript, the authors have used an impressive collection of human oocytes, immature and mature, to analyze the adaptation of the oocyte proteostasis system. Using a variety of fluorescence sensors, they show reduced lysosomal, proteasomal and mitochondrial activities in human oocytes compared to surrounding somatic cells, which correlates with a reduced abundance of proteasomal proteins and increased lysosomal exocytosis. They also report an asymmetric distribution of proteasomes and organelles during oocyte maturation.

The study is based on an impressive collection of human oocytes and shows significant changes in the proteostasis system of these cells. Although the description of proteostasis in human oocytes is interesting in itself, mechanistic insights into the adaptive mechanisms appear to be limited. The reduced capacity of the lysosomal and proteasomal systems is consistent with the previously observed long protein half-lives in these cells and is somewhat expected. Unfortunately, the manuscript does not provide any further insight into how these adaptations might be regulated. Furthermore, recently published single-cell proteomic data on human oocytes (ref 10) also showed reduced proteasome abundance, which is confirmed by the present study. Similarly, reduced mitochondrial respiratory activities have been reported previously. These considerations limit the reviewer's enthusiasm for this overall well done study.

We thank the reviewer for their constructive and positive feedback. To address their critical points, we have divided our response into subheadings:

1- Responding to *"The reduced capacity of the lysosomal and proteasomal systems is consistent with the previously observed long protein half-lives in these cells and is somewhat expected."*

Only one previous study has assessed lysosomal and proteasomal activity in mammalian (mouse) oocytes -- Zaffagnini et al., 2024 (Cell), from our lab [5]. In contrast to the reviewer's expectation, our findings in that study showed that MII-stage mouse oocytes exhibited significantly higher lysosomal and proteasomal activity than GV oocytes. This is the opposite of what we now observe in human oocytes, highlighting important species-specific

differences. Therefore, the conclusions of the current study are not only novel but also unexpected.

2- Responding to “... *Unfortunately, the manuscript does not provide any further insight into how these [proteostatic] adaptations might be regulated.*”

This was a key point of discussion with the editors prior to submitting our manuscript as a Resource. We had clarified with the editors that we “systematically investigated proteostatic mechanisms and mitochondrial activity in fertilization-ready healthy human oocytes for the first time. However, we do not yet have any mechanistic insights..” The editor then recommended submitting our manuscript as a Resource article. Due to ethical regulations, neither we nor any other research group can currently investigate these mechanisms directly in humans. As we have shown here, mouse models are no substitute as the underlying biology differs substantially from humans. We expect this paper to lay the foundation for future multi-center studies as the data it contains will provide justification for the future collection of the large sample sizes needed for mechanistic analyses, while adhering to ethical guidelines.

3- Responding to “*Furthermore, recently published single-cell proteomic data on human oocytes (ref 10) also showed reduced proteasome abundance, which is confirmed by the present study.*”

We are sorry for the confusion our wording in the manuscript generated. The referred study [22] looked into proteasomal abundance levels between GV and MII oocytes in humans and **did not observe any differences (Figure 1 of [22])**. Here, we studied the **activity** of proteasomes and lysosomes in GVs and MIIs to find that **proteasomal activity decreases during oocyte maturation** – which has not been studied neither in the cited study nor in any other study (also see new table 1). We further re-analysed the study’s raw proteomics data and found some proteasome subunits indeed decreased between GV and MII (4 subunits among 30+ proteasomal subunits in total). **Therefore, this study’s findings neither diminish nor contradict the significance of our results.**

4- Responding to “Similarly, reduced mitochondrial respiratory activities have been reported previously.”

To our knowledge, no such data exist for human GVs and MIIs using validated quantitative probes - several recent reviews also underscored this gap in the literature [23,24]. Our study is the first to directly compare mitochondrial activity across human oocyte maturation using robust methods. Until now, the literature on human oocytes has relied solely on (i) the JC-1 dye which is neither quantitative nor reliable, as it has been shown to be highly error-prone when performed without extensive controls [6], and (ii) non-physiological oocyte stages, which are metabolically distinct from in vivo isolated MII oocytes [25,26].

To summarise, while we and others have investigated this issue in model systems, no study has compared GVs, MII, and somatic cells in humans to provide insights into their respiratory status (reviewed recently in [23,24]). **This study is the first to achieve this.**

In retrospect, we realize that many of these issues above may have arisen from the reserved tone we used in the manuscript when highlighting the differences between mouse and human oocytes. **To date, no published studies have characterized proteostatic or mitochondrial activity in healthy fertilization-ready oocytes. As a result, the field has largely extrapolated findings from model organisms like the mouse.** However, our study demonstrates that this assumption does not hold: mature (MII) human oocytes exhibit markedly reduced mitochondrial, lysosomal, and proteasomal activity compared to their immature (GV) sibling oocytes—and in sharp contrast to their mouse counterparts – highlighting previously unrecognized, human-specific regulatory mechanisms in energy and waste disposal systems in oocytes. We now included a table in the manuscript (table 1) to highlight the differences between mouse and human oocyte maturation.

Additional points:

1. Although more than 100 oocytes, are used in total, the number of replicates in many analyses is limited to $N = 2 - 7$ donors (Figure 1D, Figure 2 B, C). While I understand the technical challenges, the authors should use careful wording when generalizing their findings and when referring, for example, to “large scale study...using over 100 oocytes”.

We thank the reviewer for pointing to this confusion. Now, all relevant mentions in the manuscript is more detailed:

Abstract: “Using over 100 freshly donated oocytes from 21 healthy women aged 19–34.”

Introduction: “Using a substantial sample size—over 70 MIIs and 30 GVs collected directly from 21 healthy donors—”

We’d like to highlight to the reviewer that the next comparable study used only 5 MIIs from a single donor [15]. In contrast, our use of over 100 oocytes from 21 donors represents a significant advancement over the current state of the art.

2. This also requires a careful statistical evaluation of the data. Why was an unpaired t-test used when the data are paired (donor)? It may be applicable if the variation between the oocytes is higher than between donors, but the reasoning should be explained.

We thank the reviewer for this comment. Whenever possible, we aimed to plot each oocyte independently, color-coded by donor, to represent variability both between oocytes and between donors. In principle, all the donors were young and fertile, selected according to the clinic’s oocyte donation criteria. Therefore, our goal was to highlight oocyte-to-oocyte variability within the same individual. Having said that, we agree with the reviewer that summarizing the data by donor and applying paired t-tests would provide complementary, donor-centric information that is also relevant for our conclusions.

We have therefore re-analyzed the relevant data averaging values by donor and using paired t-tests to compare them – these data are now shown in supplementary figures S2A, H-J, S3A, C-E of the revised manuscript. The reviewer will see that our conclusions remain unchanged.

The statistics in Figures 4C, F, I is missing. Figure 2B, C – missing N numbers.

Thanks for pointing out the oversight in these figures!

We have now statistically analysed the curves in Figures. 4C, F, and I, and added the N numbers in Fig. 2B, C. We thank the reviewer for highlighting this point and we apologize for the oversight. Relevant parts in methods (for linear segmentation analysis of the curves) and figure legends have been updated.

Collectively, our conclusions remain identical, and we thank the reviewer for suggesting these analyses that strongly improved our manuscript.

3. In Figure 4D, the mitochondrial mass appears higher in MII, whereas the TMRE analysis in Figure 1B shows a reduction. Although the TMRE intensity estimates “active” (e.g. membrane potential), this difference is striking. Why should the oocyte accumulate mitochondria that do not have a reduced membrane potential?

We thank the reviewer for this insightful comment and apologise for the confusion caused by not clearly referencing the relevant literature. Indeed, multiple studies across species have consistently shown that oocytes accumulate mitochondria with reduced membrane potential [1,23,27,28]. In our recent published work [1], we demonstrated that this phenomenon serves a protective function in *dormant* frog and human oocytes: by maintaining a lower membrane potential, oocytes limit mitochondrial ROS production, which is highly detrimental to oocyte quality [29–32].

Although functional data in fertilisation-ready oocytes of any species is still lacking, the prevailing hypothesis is that this ROS-avoidance mechanism remains relevant at later stages of maturation. The uncoupling between increased mitochondrial mass and reduced membrane potential in MII-stage oocytes could reflect a protective strategy—allowing oocytes to stockpile energy-generating organelles while minimising oxidative stress during the critical window prior to fertilisation.

The reviewer’s point - which is well spotted and correct - also reinforces our imaging-based conclusions: the lower TMRE labelling in MII oocytes is not due to a reduction in mitochondrial content, but rather reflects a further reduction in mitochondrial membrane potential compared to GVs (Table 1). We have now included this point in the Discussion.

“Second, while protein degradation and mitochondrial membrane potential increase during oocyte maturation in mice, both are instead downregulated in human oocytes (Fig. 1A-D). The downregulation of organelle activity in maturing human MIIs could be part of a strategy to minimize reactive oxygen species (ROS) production during the longer maturation process in humans.”

4. The manuscript would benefit from a critical assessment of limitations of the study (e.g. sample size, approaches used to assess organellar activities).

We thank the reviewer for this important suggestion. We agree that including a critical assessment of the study's limitations strengthens the manuscript. We have now added a dedicated paragraph at the beginning of the Methods section (pasted below for reviewer's convenience) acknowledging key limitations. If accepted, we will also consult the editor about the possibility of adding a stand-alone "Limitations of the Study" section after the Discussion.

"A few limitations of this study should be acknowledged. While this study includes, to our knowledge, the largest number of human oocytes analysed for organellar activity to date, the sample size is necessarily constrained by the limited availability and ethical considerations surrounding the use of healthy human oocytes. However, the relatively large number of oocytes in our study enabled paired analyses of sibling oocytes, strengthening the internal validity of our comparisons. That said, including more oocytes per donor or increasing the number of donors could provide greater statistical power to support the conclusions of this manuscript. In addition, our assessment of organellar function—specifically mitochondrial membrane potential, lysosomal and proteasomal activity—relied on live-cell fluorescent probes. Although these are widely accepted and commonly used tools in the field, they do not provide mechanistic insights. Future studies incorporating complementary functional assays that would involve multi-centre studies to increase oocyte numbers will be essential to further elucidate the molecular mechanisms underlying organelle regulation during oocyte maturation. Despite these limitations, our findings provide a robust and novel resource for understanding subcellular changes in human oocytes during maturation."

References:

1. Rodríguez-Nuevo, A. *et al.* (2022) Oocytes maintain ROS-free mitochondrial metabolism by suppressing complex I. *Nature* DOI: 10.1038/s41586-022-04979-5
2. Rayon, T. *et al.* (2020) Species-specific pace of development is associated with differences in protein stability. *Science* 369
3. Nakanoh, S. *et al.* (2024) Protein degradation shapes developmental tempo in mouse and human neural progenitors. *bioRxiv* DOI: 10.1101/2024.08.01.604391
4. Matsuda, M. *et al.* (2024) Systematic analysis of protein stability associated with species-specific developmental tempo. *bioRxiv* DOI: 10.1101/2024.06.07.597977
5. Zaffagnini, G. *et al.* (2024) Mouse oocytes sequester aggregated proteins in degradative super-organelles. *Cell* DOI: 10.1016/j.cell.2024.01.031
6. AL-Zubaidi, U. *et al.* (2019) The spatio-temporal dynamics of mitochondrial membrane potential during oocyte maturation. *Mol Hum Reprod* 25, 695–705
7. Cheng, S. *et al.* (2022) Mammalian oocytes store mRNAs in a mitochondria-associated

membraneless compartment. *Science* 378

8. Satouh, Y. et al. (2024) Endosomal-lysosomal organellar assembly (ELYSA) structures coordinate lysosomal degradation systems through mammalian oocyte-to-embryo transition. DOI: 10.7554/elife.99358.1

9. Harasimov, K. et al. (2024) The maintenance of oocytes in the mammalian ovary involves extreme protein longevity. *Nat. Cell Biol.* DOI: 10.1038/s41556-024-01442-7

10. Das, M. and Son, W.-Y. (2023) In vitro maturation (IVM) of human immature oocytes: is it still relevant? *Reprod. Biol. Endocrinol.* 21, 110

11. So, C. et al. (2019) A liquid-like spindle domain promotes acentrosomal spindle assembly in mammalian oocytes. *Science* 364, eaat9557

12. Holubcova, Z. et al. (2015) Error-prone chromosome-mediated spindle assembly favors chromosome segregation defects in human oocytes. *Science* 348, 1143–1147

13. Trebichalská, Z. et al. (2020) Cytoplasmic maturation in human oocytes: an ultrastructural study. *Biol. Reprod.* 104, ioaa174-

14. Wu, T. et al. (2022) The mechanism of acentrosomal spindle assembly in human oocytes. *Science* 378

15. Santos, T. et al. (2024) Stereological study of organelle distribution in human mature oocytes. *Sci. Rep.* 14, 25816

16. ART, T.E.W.G. on U. in et al. (2019) Recommendations for good practice in ultrasound: oocyte pick up. *Hum. Reprod. Open* 2019, hoz025

17. McCulloh, D.H. et al. (2020) Follicle size indicates oocyte maturity and blastocyst formation but not blastocyst euploidy following controlled ovarian hyperstimulation of oocyte donors. *Hum. Reprod.* 35, 545–556

18. Pors, S.E. et al. (2022) Oocyte diameter predicts the maturation rate of human immature oocytes collected ex vivo. *J. Assist. Reprod. Genet.* 39, 2209–2214

19. Combelles, C.M.H. et al. (2002) Assessment of nuclear and cytoplasmic maturation in in-vitro matured human oocytes. *Hum. Reprod.* 17, 1006–1016

20. Salimov, D. et al. (2023) Chromatin Morphology in Human Germinal Vesicle Oocytes and Their Competence to Mature in Stimulated Cycles. *Cells* 12, 1976

21. Tatičková, M. et al. (2023) The ultrastructural nature of human oocytes' cytoplasmic abnormalities and the role of cytoskeleton dysfunction. *FS Sci.* 4, 267–278

22. Galatidou, S. et al. (2024) Single-cell proteomics reveals decreased abundance of proteostasis and meiosis proteins in advanced maternal age oocytes. *Mol. Hum. Reprod.* 30, gaae023

23. Bahety, D. et al. (2024) Mitochondrial morphology, distribution and activity during oocyte development. *Trends Endocrinol. Metab.* DOI: 10.1016/j.tem.2024.03.002

24. Yildirim, R.M. and Seli, E. (2024) The role of mitochondrial dynamics in oocyte and early embryo development. *Semin. Cell Dev. Biol.* 159, 52–61

25. Wilding, M. et al. (2001) Mitochondrial aggregation patterns and activity in human oocytes and preimplantation embryos. *Hum. Reprod.* 16, 909–917

26. Blerkom, J.V. and Davis, P. (2006) High-polarized ($\Delta\Psi$ mHIGH) mitochondria are spatially polarized in human oocytes and early embryos in stable subplasmalemmal domains: developmental significance and the concept of vanguard mitochondria. *Reprod. Biomed. Online* 13, 246–254

27. Ferrer-Vaquero, A. et al. (2019) Altered cytoplasmic maturation in rescued in vitro matured oocytes. *Hum. Reprod.* 34, 1095–1105
28. Adhikari, D. et al. (2022) Oocyte mitochondrial key regulators of oocyte function and potential therapeutic targets for improving fertility. *Biol. Reprod.* 106, 366–377
29. Aitken, R.J. (2020) Impact of oxidative stress on male and female germ cells: implications for fertility. *Reproduction* 159, R189–R201
30. Agarwal, A. et al. (2005) Role of oxidative stress in female reproduction. *Reprod. Biol. Endocrinol.* 3, 28
31. Prasad, S. et al. (2016) Impact of stress on oocyte quality and reproductive outcome. *J. Biomed. Sci.* 23, 36
32. Almansa-Ordonez, A. et al. (2020) Oxidative Stress in Reproduction: A Mitochondrial Perspective. *Biology* 9, 269

Dear Elvan,

Thank you for submitting your manuscript together with the reviews from another journal and your point-by-point response to them to The EMBO Journal. I have now received input on the revised study from both original reviewers. As you can see, they are satisfied with the revisions and now recommend publication of your revised study in our journal. Therefore, I will accept your manuscript after reformatting of the manuscript according to The EMBO Journal guidelines as listed below.

1. Please provide up to five keywords.
2. Please submit a complete author checklist, which you can download from our author guidelines (<https://www.embopress.org/pb-assets/embo-site/EMBO%20Press%20Author%20Checklist-1642513524327.xlsx>). Please insert information in the checklist that is also reflected in the manuscript. The completed author checklist will also be part of the Review Process File.
3. Please upload the main and supplementary (Expanded View) figures as individual production quality figure files in the .eps, .tif, or .jpg format (one file per figure).
 1. Supplementary figures should be renamed into Expanded View (EV) figures. EV Figures should be cited as 'Figure EV1, Figure EV2' etc. in the text and their respective legends should be included in the main text after the legends of regular figures. Further information on the format is available here: <https://www.embopress.org/page/journal/14602075/authorguide#expandedview>.
4. Please make sure that the order of the sections in the manuscript is as follows: abstract, introduction, results, discussion, materials & methods, data availability section, acknowledgments, disclosure statement and competing interests, references, main figure legends, tables, expanded figure legends.
5. Please check that the funding information is correct and identical both in the manuscript and our online system. Currently, the Spanish Ministry of Science and Innovation through the Centro de Excelencia Severo Ochoa (CEX2020-001049-S, MCIN/AEI /10.13039/501100011033), and the Generalitat de Catalunya CERCA programme are missing in our online system.
6. At EMBO Press we ask authors to provide source data for the main manuscript figures. You will receive a separate email with instructions for providing source data with your revised manuscript, including how to upload and organize the files.
7. Please rename "Competing Interests" section into "Disclosure and competing interests statement" (further info: <https://www.embopress.org/page/journal/14602075/authorguide#conflictsofinterest>).
8. Please update references according to The EMBO Journal style - they should be listed in alphabetical order. Where there are more than 10 authors on a paper, the first 10 should be listed, followed by 'et al.' Please see further information here: <https://www.embopress.org/page/journal/14602075/authorguide#referencesformat>
9. All Materials and Methods need to be described in the main text using our 'Structured Methods' format. According to this format, the Methods section includes a Reagents and Tools Table (listing key reagents, experimental models, software and relevant equipment and including their sources and relevant identifiers) followed by a Methods and Protocols section describing the methods, ideally using a step-by-step protocol format. The aim is to facilitate adoption of the methodologies across labs. Please download and fill our Reagents and Tools Table template (.docx), which you can find in our author guidelines: <https://www.embopress.org/page/journal/14602075/authorguide#structuredmethods>
An example of a Method paper with Structured Methods can be found here: <https://www.embopress.org/doi/10.15252/msb.20178071>.
- When submitting your revised manuscript, please upload it as a separate file choosing the file type "Reagent Table". The information currently provided in Key Resources Table could be adapted to this format.
10. Please rename Table S1 into Table EV1 and update the callouts throughout the text. Please add the title and legend to the top of the page.
11. Figure S4, panel label "A" can be removed, as it contains a single panel.
12. Please rename the "Code availability" section into "Data availability", while the note about the availability of raw data can be removed from this section, as the source data will be linked to your manuscript upon publication. More information on the format of this section can be found here: <https://www.embopress.org/page/journal/14602075/authorguide#dataavailability>.
13. During our routine image quality check, significant similarities were found between figures 4G and S5A (GV, TMRE). Please check - are the images derived from the same oocyte before and after treatment with Verapamil?
14. Figure panels 3A (upper row, proteost.) and S1D (confocal panels for the upper row) are devoid of signal. I appreciate that these are negative controls, however, some level of background signal would be expected. Please provide source data for these panels.
15. Our data editors have flagged the following issues in figure legends that need correcting:
 - Please provide the exact p values in the legends of figures 1B, E, F, G; S2 I, J.
 - Please define the box plots in terms of minima, maxima, centre, bounds of box and whiskers, and percentile in the legends of figures 3B, S1 A.
 - Please provide information on the number and nature of replicates in the legend of figure 3B.
16. Papers published in The EMBO Journal are accompanied online by a 'Synopsis' to enhance discoverability of the manuscript. Please submit a short (1-2 sentences) summary of the findings and their significance in addition to the already provided bullet points highlighting the key results. Please also send us a synopsis image that is 550x300-600 pixels large (width x height, jpeg or png format). You can either show a model or key data in the synopsis image. Please note that the image size is

rather small and that text needs to be readable at the final size.

With best wishes,

Ieva

We realize that it is difficult to revise to a specific deadline. In the interest of protecting the conceptual advance provided by the work, we recommend a revision within 3 months (26th Aug 2025). Please discuss the revision progress ahead of this time with the editor if you require more time to complete the revisions. Use the link below to submit your revision:

Referee #1:

The authors highlighted the novelty of the study in a more comprehensive manner and addressed the limitations of the study in the discussion. Most importantly, the statistical analysis and its description has been improved.

Referee #2:

This is a review of a revision that was transferred. The authors have addressed my concerns carefully and thoughtfully. I appreciate their efforts on addressing these concerns. I am satisfied with their edits.

All editorial and formatting issues were resolved by the authors.

Dear Elvan,

Thank you for addressing the final editorial points. I am now pleased to inform you that your manuscript has been accepted for publication in the EMBO Journal.

Before we forward your manuscript to our publishers, we would like to propose some edits in the manuscript abstract and synopsis (please see below and the attached text file). I have also written a short blurb that will accompany the title of your manuscript in our online system. Please let me know if any corrections or adjustments are needed.

Blurb:

In contrast to the mouse, mature human oocytes show reduced activity of lysosomes, proteasomes and mitochondria, highlighting species-specific differences in reproductive biology.

Synopsis:

Oocytes are long-lived cells that require specialized adaptations to maintain their fitness. This study shows that human oocytes display low lysosomal, proteasomal, and mitochondrial activity, which declines further during oocyte maturation and likely protects oocyte integrity during prolonged developmental arrest.

- Immature oocytes show higher proteolytic activity than mature oocytes.
- Mature oocytes have a reduced number of lysosomes and reduced mitochondrial membrane potential.
- Human oocytes contain aggregated proteins within a few enlarged lysosomes distributed throughout the cytoplasm.
- Oocyte maturation is accompanied by rearrangement of mitochondria and proteasomes from the perinuclear area to the cytoplasm.

If you have any questions, please do not hesitate to contact the Editorial Office. Thank you for your contribution to The EMBO Journal, and congratulations on a nice study!

With best wishes,

Ieva
